# Transit amplifying cells coordinate mouse incisor mesenchymal stem cell activation

Jemma Victoria Walker [1], Heng Zhuang [2,3], Donald Singer [1], Charlotte Sara Illsley [1], Wai Ling Kok [1], Kishor K. Sivaraj [4], Yan Gao [1,5], Chloe Bolton [1], Yuying Liu [1,6], Mengyuan Zhao [7], Portia Rebecca Clare Grayson [1], Shuang Wang [8], Jana Karbanová [9], Tim Lee [10], Stefano Ardu [11], Qingguo Lai [12,13], Jihui Liu [6], Moustapha Kassem [14,15], Shuo Chen [16], Kai Yang [5], Yuxing Bai [5], Christopher Tredwin [1], Alexander C. Zambon [10], Denis Corbeil [9], Ralf Adams [4], Basem M. Abdallah [14,17] & Bing Hu [1]

Stem cells (SCs) receive inductive cues from the surrounding microenvironment and cells. Limited molecular evidence has connected tissue-specific mesenchymal stem cells (MSCs) with mesenchymal transit amplifying cells (MTACs). Using mouse incisor as the model, we discover a population of MSCs neibouring to the MTACs and epithelial SCs. With *Notch* signaling as the key regulator, we disclose molecular proof and lineage tracing evidence showing the distinct MSCs contribute to incisor MTACs and the other mesenchymal cell lineages. MTACs can feedback and regulate the homeostasis and activation of CL-MSCs through Delta-like 1 homolog (Dlk1), which balances MSCs-MTACs number and the lineage differentiation. *Dlk1*'s function on SCs priming and self-renewal depends on its biological forms and its gene expression is under dynamic epigenetic control. Our findings can be validated in clinical samples and applied to accelerate tooth wound healing, providing an intriguing insight of how to direct SCs towards tissue regeneration.

[1] Stem Cells & Regenerative Medicine Laboratory, Peninsula Dental School, University of Plymouth, 16 Research Way, Plymouth PL6 8BU, UK. [2] Department of Cariology, Endodontology and Operative Dentistry, Peking University School and Hospital of Stomatology, 22 South Zhongguancun Avenue, Haidian District, 100081 Beijing, China. [3] Taikang Bybo Dental Group Beijing, No. 4 Building, 18 Qinian Avenue, Dong Cheng District, 100062 Beijing, China. [4] Max Planck Institute for Molecular Biomedicine, Department of Tissue Morphogenesis and University of Münster, Faculty of Medicine, 48149 Münster, Germany. [5] Department of Orthodontics, School of Stomatology, Capital Medical University, 4 Tian Tan Xi Li, Dong Cheng District, 100050 Beijing, China. [6] Department of Orthodontics, Shenyang Stomatological Hospital, 38 Zhong Shan Road, He Ping District, 110002 Shen Yang, China. [7] Institute of Dental Research, Beijing Stomatological Hospital, Capital Medical University, 4 Tian Tan Xi Li, Dong Cheng District, 10050 Beijing, China. [8] Faculty of Dentistry, National University of Singapore, 11 Lower Kent Ridge Road, Singapore 119083, Singapore. [9] Tissue Engineering Laboratories (BIOTEC), Technische Universität Dresden, 01307 Dresden, Germany. [10] Department of Biopharmaceutical Sciences, Keck Graduate Institute, 535 Watson Drive, Claremont, CA 91711, USA. [11] Clinique Universitaire de Médecine Dentaire, Université de Genève, 19 rue Lombard, Geneva CH-1206, Switzerland. [12] Department of Oral and Maxillofacial Surgery, the Second Hospital of Shandong University, 247 Beiyuan Street, 250033 Jinan, China. [13] Research Center of 3D Printing in Stomatology of Shandong University, 247 Beiyuan Street, 250033 Jinan, China. [14] Molecular Endocrinology Lab (KMEB), Department of Endocrinology, Odense University Hospital & University of Southern Denmark, 5000 Odense, Denmark. [15] Department of Cellular and Molecular Medicine, DanStem (Danish Stem Cell Center), Panum Institute, University of Copenhagen, 2200 Copenhagen, Denmark. [16] Department of Developmental Dentistry, Dental School, The University of Texas Health Science Center at San Antonio, San Antonio, TX 78229, USA. [17] Department of Biological Sciences, College of Science, King Faisal University, Hofuf 11533, Saudi Arabia. [18] The two authors contribute equally: Jemma Victoria Walker, Heng Zhuang. Correspondence and requests for materials should be addressed to B.H. (email: bing.hu@plymouth.ac.uk)

Tissue development and regeneration largely rely on stem cells (SCs) that give rise to multiple cell lineages. In a healthy tissue, proper activation of SCs and preservation of SC reservoir are rigidly balanced to ensure that accurate SC homeostasis and differentiation are in place[1,2]. In most cases, SCs reside in a specialized and dynamic microenvironment, also called the SC niche, and receive feedback cues notably from their surrounding environment including those from neighboring heterologous cell populations[3]. The signaling that maintains SC niche have been thoroughly investigated. Both Wnt and Notch pathways have been implicated in regulation of SCs within muscle[4], intestinal epithelium[5], and interfollicular epidermis[6]. Various cell types or even terminally differentiated cells re able to provide feedback loops to SCs. For instance, in the intestinal epithelium, Paneth cells play a key role in SC niche determination[5]. In hair follicle keratinocytes, the transit amplifying cells (TACs) can orchestrate SC activity and tissue regeneration through the Sonic Hedgehog (SHH) pathway[7].

TACs are committed and proliferative direct lineage of primed SCs[8]. For most of the mammalian organs, upon the completion of development or regeneration, the TACs vanish and tissue growth stops[9]. TACs induction, maintenance, and differentiation therefore must be rigorously controlled and directed by molecular signaling and cellular microenvironment. In the case of regeneration, SCs can also be re-activated to produce new TACs when the tissues of interest receive stimulating (activating) signals, such as molecular cues for regeneration released by wounded tissues[10]. To date, most research focuses on epithelial tissues, such as intestinal epithelium, which have lifelong persistent TACs[5,11], while in hair follicles, keratinocytes contain periodic TACs that reenter the cell cycle[7]. Discovering how the signaling pathways control TACs to SCs communications is challenging, partially due to the fact that most of tissues do not possess sufficient amounts of SCs, TACs, and differentiated cell populations at the same time, and importantly in close vicinity. However, in the tissues involved in epithelial–mesenchymal interactions, while the epithelial SCs and TACs are often investigated, the neighboring mesenchymal compartment's SCs and niche have been less documented.

Mammalian tooth development involves dynamic epithelial–mesenchymal interactions[12]. The mouse incisors undergo lifelong growth as a result of epithelial–mesenchymal interactions. Each mouse incisor tooth has a persisting epithelial SC-TAC zone at its posterior end of the tooth, named cervical loop (CL) based on the epithelial structure (Fig. 1a)[13]. Specifically, the CL epithelium has a defined SC niche located at the middle of the enlarged head like structure (Fig. 1b) where cells highly express markers such as Sox2, Bmi1, and Lgr5[14–16], while the neighboring shoulder and upper front body epithelium admit TACs (Supplementary Fig. 1a). The latter are highly positive for Ki67 and the other markers such as Prom1 and SHH[17,18]. Next to the epithelial SCs, TACs form a cluster of Ki67-positive cells that connect the SCs with differentiating cells named pre-ameloblasts, the cells start polarizing and deposit enamel matrix at the epithelial–mesenchymal junction (Fig. 1a, c). Facing the epithelium TACs, the incisor mesenchyme also has a distinct mesenchymal TAC (MTAC) zone that the cells highly express Ki67 (Fig. 1c), growth factors such as Fgf3/7/10 (refs. [19,20]), and epigenetic repressors such as Ring1a and 1b[19].

Besides the population of MTACs, recent evidence has revealed a mesenchymal SCs (MSCs) niche around the perineurovascular bundle (NVB) region[21,22] where cells have been found to be label retaining and potentially with Glia origin as they are positive for Plp promoter-driven Cre[21], and Gli1 Cre and Thy-1 Cre[22,23]. The NVB-MSCs can release progenitors that participate in tissue repair and replenish the undifferentiated mesenchyme upon injury[21–23]. However, little experimental evidence has been shown that the MTACs are direct derivatives of the NVB-MSCs. Furthermore, the NVB SC niche is not able to contribute all of the differentiated cells in the incisor pulp[21–23]. Together this hints at the co-existence of another of MSC population(s) contributing as the precursors of MTACs. Determining the existence of such population of MSCs, and potentially a distinct niche, remains a challenge, as does defining how MTACs could be maintained and communicate with MSCs within the mouse incisor.

In the present study, we discover an MSC population associating with the mouse incisor CL region and show that the MTACs can feedback to and control those MSCs through the Dlk1, a Notch pathway ligand, which is important in inducing MSCs to MTACs transition, lineage differentiation, and tissue regeneration, a mechanism that can potentially be translated into regenerative medicine.

## Results

**Mouse incisor tooth harbors a distinct MSC population.** Given the mesenchymal cells contacting the opposing CL epithelial SCs were also positive for Gli1 Cre[17,22] and essentially those cells were immediate neighbors of the MTACs (Fig. 1c), we asked whether they represented a special population of mesenchymal cells. We first performed immunostaining using specific antibodies against Gli1 (ref. [22]), Thy-1 (ref. [23]) and PDGFrβ on postnatal day 7 (P7) CD1 mouse lower incisors and found that they were highly expressed in those cells, as in the NVB-MSCs (Fig. 1c, Supplementary Fig. 1a). Unlike the NVB-MSCs, this population of mesenchymal cells was negative for Sca1 and CD106 (Supplementary Fig. 1a). These observations could be further confirmed by micro-dissection followed by flow cytometry analysis (Supplementary Fig. 1b). We next profiled the expression of a panel of classical MSC and TAC markers[24], using laser capture microdissection (LCM; Supplementary Fig. 1c) and real-time RT-PCR (Fig. 1d, e). The results showed that indeed the population of mesenchymal cells highly expressed most of the general MSCs markers in mRNA levels, such as Ccnd3, Cdkn1a, Smarca2, and Zbtb20 (Fig. 1d). Meanwhile the MTACs displayed a group of distinctly different markers such as Anln, Ccna2, and Top2a (Fig. 1e). Conversely, NVB-MSCs expressed most of the MSC, and also MTAC markers (Fig. 1d, e), therefore, could not be molecularly dissected from MTACs based on mRNA expression (Fig. 1e). We thus named the MSCs population as CL-MSCs, in comparison to the NVB-MSCs.

We next performed additional immunofluorescent analysis and confirmed that SmarcA2 and Zbtb20 were two robust CL-MSCs markers (Fig. 1f). Essentially, SmarcA2 showed transitional expression overlapping with Ki67 at the junction of CL-MSCs and MTACs regions (Fig. 1g), which further hinted the linkage between CL-MSCs and MTACs. Similar to the flow cytometry data, CL-MSCs were found positive for CD73 and CD29 by immunofluorescent analysis (Supplementary Fig. 1d).

It has been reported that NVB-MSCs have a critical role in incisor homeostasis[22]. To evaluate which MSC population could support the growth of CL, we performed CL organ culture (Fig. 2a). Interestingly, we found the NVB structures as observed by CD106 and Neural Filament staining had a short life in vitro, i.e. less than 3 days (Fig. 2b, c). Quite the contrary, the incisor CL-MSCs still persisted and expressed markers such as Thy-1 (Fig. 2d), and connected to the MTACs that are positive to Ki67 (Fig. 2e), therefore, contributed to the CL growth ex vivo (Fig. 2a–e).

**CL-MSCs are multipotent progenitors of odontoblasts.** To explore if CL-MSCs were the progenitors of odontoblasts, the

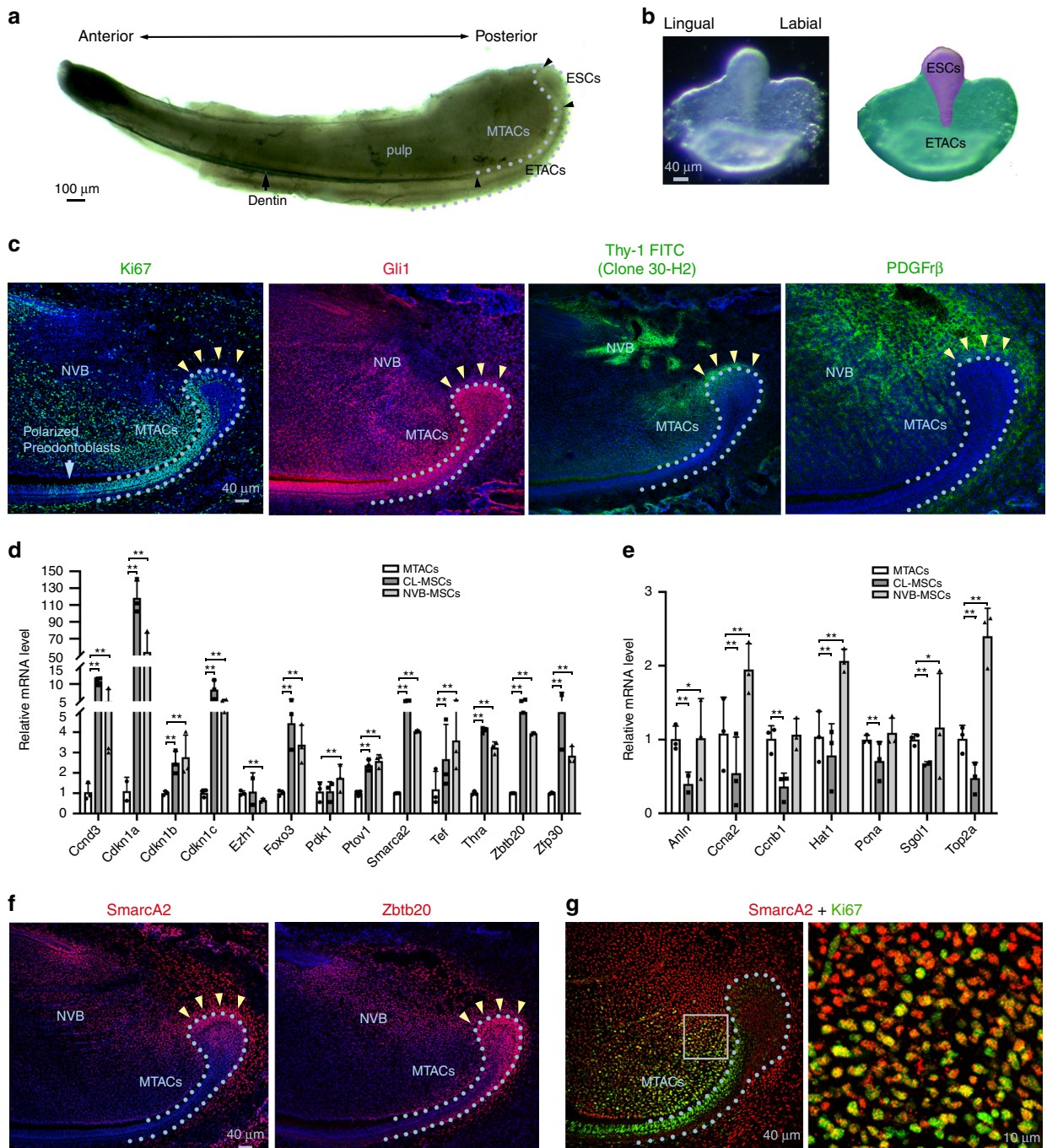

**Fig. 1** Identification of the CL-MSCs in the mouse incisor tooth. **a** Whole view of one P7 mouse lower incisor. Black arrowheads show approx. boundaries of indicated epithelial cells zones: ESCs epithelial stem cells, ETACs epithelial transit amplifying cells, MTACs, mesenchymal transit amplifying cells, and dentin layer. Light blue dotted line indicates epithelial–mesenchymal junction. **b** Stereo front view of a P7 incisor CL epithelium (left) and arbitrarily colored indexes for the ESCs (purple) and ETACs (green) regions (right). **c** Immunofluorescence analysis of indicated antigenic markers in P7 incisors. Nuclei were counterstained with DAPI. Light blue dotted line shows epithelial–mesenchymal junction while yellow arrowheads indicate the mesenchymal cells named CL-MSCs. NVB neurovascular bundle region. **d**, **e** Gene expression profiling using real time RT-PCR analysis on laser captured MTACs, CL-MSCs, and NVB-MSCs from $n = 5$ biologically independent animals, using two panels of general MSCs (**d**) and MTACs (**e**) markers. Triplicated samples were used for each gene. Error bars represent standard deviation. Statistical analysis was performed with two-way ANOVA followed by Bonferoni correction. No asterisk: $p > 0.05$; one asterisk: $p < 0.05$; two asterisks: $p < 0.01$. **f**, **g** Immunofluorescence analysis using indicated markers. In **g**, squared region in the right panel is enlarged in the left one. Note the overlapped expression of SmarcA2 and Ki67 at the junction of CL-MSCs and MTACs. Light blue dotted line shows epithelial–mesenchymal junction while yellow arrowheads indicate the mesenchymal cells named CL-MSCs. Bars: **a**: 100 μm; **b**, **c**, **f**, **g** (left panel): 40 μm; **g** (right panel): 10 μm

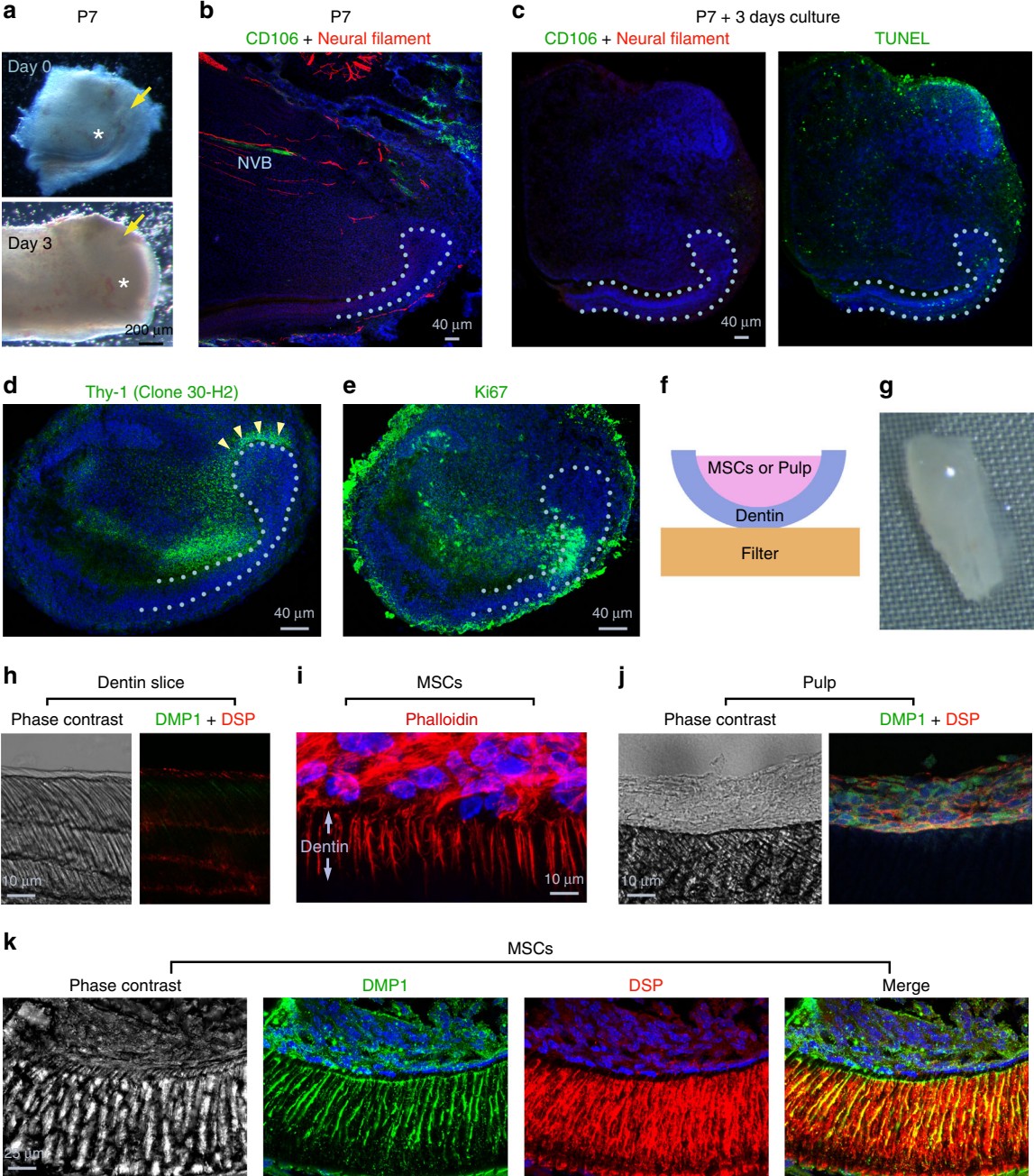

**Fig. 2** CL-MSCs contribute to mouse incisor tooth growth and odontoblast differentiation. **a** Representative micrographs for a P7 incisor CL before (day 0) and after 3 days of culture (Day 3). Note the significant expansion of MTACs region after 3 days (white asterisks). Yellow arrows indicate CLs. **b** Immunofluorescence analysis of indicated markers in the P7 CL region. Nuclei were counterstained with DAPI. Light blue dotted line shows epithelial–mesenchymal junction. NVB neurovascular bundle region. **c–e** Immunofluorescence analysis and TUNEL assay of indicated markers in the cultured CL as shown in **a**. TUNEL was performed on the same sample showed in **c** after CD106 and Neural Filament staining and exposure to light to bleach the staining. Light blue dotted line shows epithelial–mesenchymal junction. **f** Schematic indication of in vitro dentin slice-based differentiation assay model. **g** Stereo view of a recombined dentin slice with a fresh CL-MSC tissue. **h** Phase contrast pictures of decellular dentin slice (left), and DMP1 and DSP immunostaining on the slice (right). **i** CL-MSCs cultured on dentin slice for 4 days and stained with phalloidin-Alexa 568 (actin) and DAPI (nucleus). **j**, **k** Pulp cells (**j**) or MSCs (**k**) were combined with dentin slices and cultured for 7 days prior the immunostaining for DMP1 and DSP. Nuclei were counterstained with DAPI. Corresponding phase contrast images are shown. Note while DMP1 and DSP remained within tooth pulp cell layer, numerous cellular processes are invading dentin slice when MSCs are used. Bars: **a**: 200 μm; **b–e**: 40 μm; **h–j**: 10 μm; **k**: 25 μm

differentiated tooth mesenchymal cells responsible for dentin formation, we performed chimeric-like tissue culture by recombining freshly isolated CL-MSCs with devitalized dentin slice (Fig. 2f, g)[25]. We observed a rapid polarization of the CL-MSCs by producing branched protrusions into the dentin slices after only 4 days in culture (Fig. 2h, i). At day 7, while the tooth pulp

cells only retained on the surface of the dentin slices (Fig. 2j), the CL-MSC-derived cells fully polarized and secreted Dentin Sialo Protein (DSP) and Dentin Matrix Protein 1 (DMP1) (Fig. 2k).

The application of SCs in tissue engineering and regenerative medicine often requires overcoming the major challenge of amplifying the cells in vitro without losing their pluripotency and

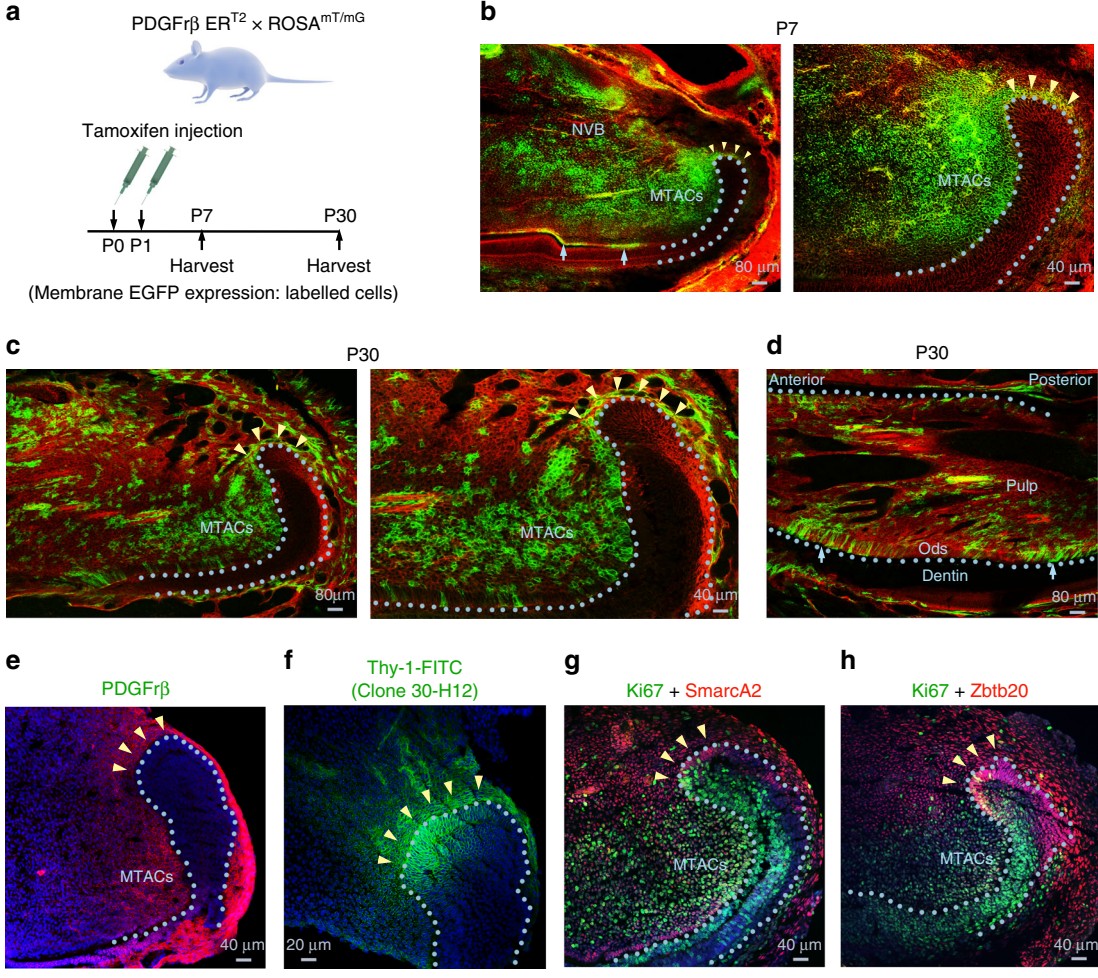

**Fig. 3** Lineage tracing MSCs in the incisor tooth. **a** Strategies of *PDGFrβ Cre ER*T2 and *ROSA* mT/mG mouse crossing and tamoxifen injection and sample harvesting are displayed. Illustration taken from https://www.somersault1824.com/science-illustrations/, which provides png images free to use and alter under the creative commons 4.0 license. **b** Representative mT/mG signal analysis at P7. Arrows mark clusters of odontoblasts that are EGFP positive. MTACs mesenchymal transit amplifying cells, NVB neurovascular bundle region. **c, d** Representative mT/mG signal analysis at P30. **e–h** Immunofluorescence analysis on the specific markers on P30 CD1 lower incisors. Nuclei were counterstained with DAPI. Yellow arrows mark clusters of odontoblasts that were EGFP positive. Light blue dotted lines indicate epithelial–mesenchymal junctions. Ods odontoblasts. Bars: **b** (left panel), **c** (left panel), and **d**: 80 μm; **b** (right panel), **c** (right panel), and **e, g, h**: 40 μm; **f**: 20 μm

differentiation capabilities. We established four independent lines of primary CL-MSCs and maintained them in culture in vitro for up to four passages using defined culture conditions (Supplementary Fig. 2a, b). Cultured CL-MSCs maintained MSC marker expression (Supplementary Fig. 2c). Cells responded to nutrient starvation and replenishment conditions by expressing MSC and MTAC markers, respectively (Supplementary Fig. 2d). Importantly, when co-cultured with dentin slices under differentiation conditions, even the passage 4 cells could give rise to odontoblast-like cells by forming protrusions into the dentin tubes and secreting DSP and DMP1 (Supplementary Fig. 2e). Additionally, we could confirm that in vitro, the CL-MSCs could also differentiate into adipocyte-, osteoblast-, and chondrocyte-like cells under proper differentiation conditions (Supplementary Fig. 3).

To follow the CL-MSCs fate, we traced and differentiated incisor MSCs. Based on our molecular marker profiling (Fig. 1c, Supplementary Fig. 1a), we adopted a lineage tracing strategy by crossing *PDGFrβ Cre ER*T2 mice[26] with *ROSA* mT/mG mice[27] (Fig. 3a). By injecting tamoxifen at P0 and 1, we could then trace the labeled cells persisting after birth. At P7, we found that most of the CL-MSCs and MTACs were labeled, and continuous labeling along CL-MSC and MTAC axis was also observed (Fig. 3b). Clusters of labeled cells were also detected within the odontoblast layer (Fig. 3b). At P30, when incisors have already erupted and functioning, the labeling was still persisting in the CL-MSCs, and again remained positive up to MTACs (Fig. 3c). It is noticeable that at P30 in the CL epithelial–mesenchymal junction, there was still an abundance of cells at the MTAC region (Fig. 3c) as well as in the odontoblast layer (Fig. 3c, d). Consistent with the findings and similar to P7, CL-MSCs still express MSC markers such as PDGFrb, Thy-1, SmarcA2, and Zbtb20 at P30 (Fig. 3e–h). Altogether our data suggest that the population of the mesenchymal cells is indeed a distinct MSC population, which clearly contrasts to the MTACs but are different from the NVB-MSCs.

**CL-MSCs and MTACs have contrasted Notch pathway profiles.** *Notch* is a key signaling pathway in SC fate determination and niche maintenance in a number of biological systems such as nerve[28], muscle[29], mammary gland epidermis[30], and inter-follicular epidermis[31]. In murine incisor tooth, little information has been documented about *Notch* signaling activity although some data have revealed the expression of *Notch1, 2,* and *3*

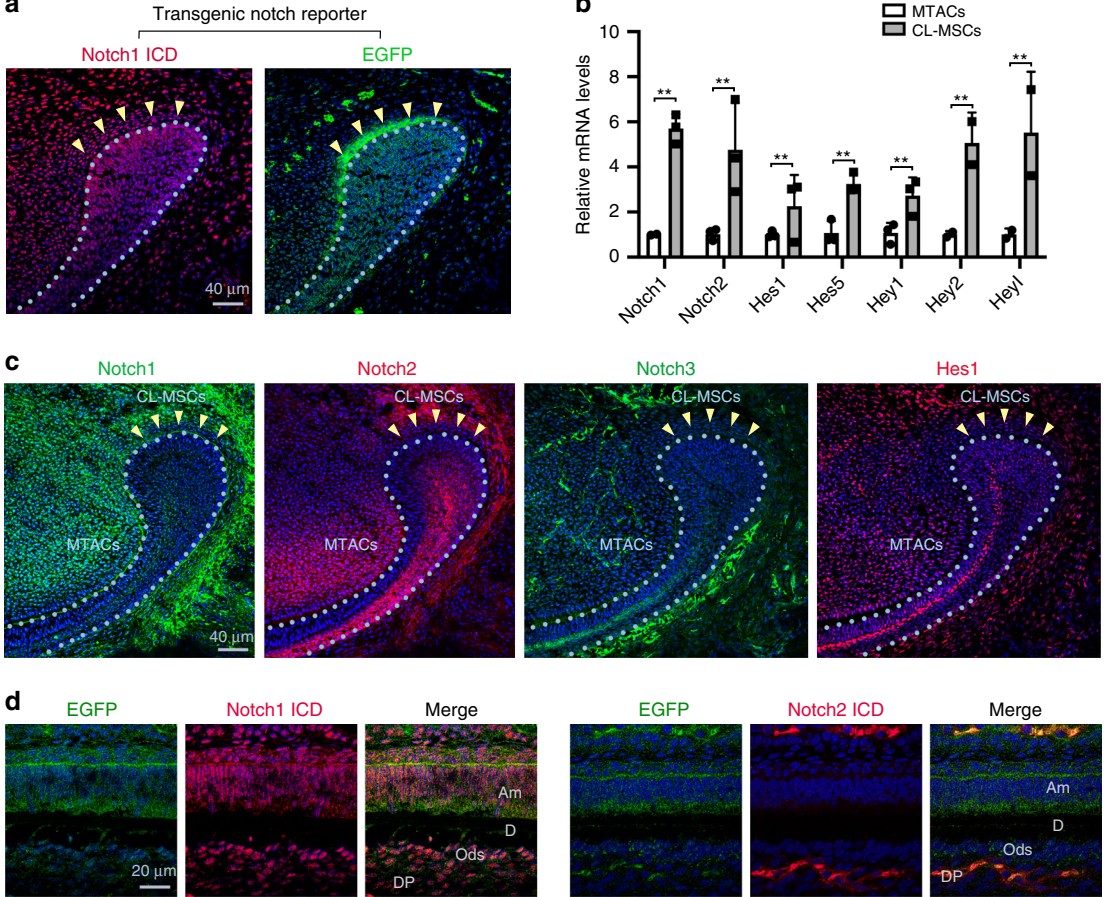

**Fig. 4** *Notch* pathway is dynamically modulated in mouse incisor MSCs and TACs. **a** Double staining of Notch1 ICD and EGFP on lower incisor CL region of P5 transgenic TNR mice. Nuclei were counterstained with DAPI. **b** Real-time RT-PCR profiling of mRNA expression of the Notch receptors and downstream targets for MTACs and CL-MSCs from $n = 5$ biologically independent animals. Triplicated samples were used for each gene. Error bars represent standard deviation. Statistical analysis was performed with two-way ANOVA followed by Bonferoni correction. **p < 0.01. **c** Immunofluorescence analysis of Notch 1–3 and Hes1 in the P7 incisor CL region. Nuclei were counterstained with DAPI. **d** Double staining of EGFP either with Notch1 ICD (left) or Notch2 ICD (right) on the differentiated labial enamel–dentin junction. Am ameloblasts, D dentin, DP dental pulp, Ods odontoblasts. Nuclei were counterstained with DAPI. Yellow arrows mark clusters of odontoblasts that were EGFP positive. Light blue dotted lines indicate epithelial–mesenchymal junctions. Bars: **a** and **c**: 40 μm; **d**: 20 μm

mRNA therein[32]. We therefore first evaluated the *Notch* activity in our system. Using *Transgenic Notch Reporter* (*TNR*) mice where enhanced green fluorescent protein (EGFP) is driven by four RBP-Jkappa binding sites and a minimal SV40 promoter[33] and double immunofluorescent staining with an antibody directed against intracellular domain (ICD) of Notch1, we found that CL-MSC region harbored a high level of Notch signaling activity where a thin layer of cells located about 20 μm above the CL epithelial SC region was Notch1 ICD/EGFP-double positive (Fig. 4a). The MTAC region showed also positivity, yet weaker (Fig. 4a). Consistently, the mRNA expression of *Notch1* and *Notch2* gene products as well as their downstream molecular targets, i.e. *Hes1, Hes5, Hey1, Hey2*, and *Heyl* were significantly higher in the CL-MSCs than the MTACs as observed by real-time RT-PCR profiling (Fig. 4b). In contrast, Notch 2 ICD and Hes1 protein expression were found higher in MTACs than CL-MSCs (Fig. 4c), while Notch1 and its ICD expressions did not show major differences (Fig. 4a, c). The expression of Notch3 was restricted to endothelial cells present in the mesenchyme (Fig. 4c). Finally, the analysis of labial enamel–dentin junction in the tooth pulp revealed that Notch1 ICD was highly expressed in differentiated odontoblasts while Notch2 ICD was higher in the adjacent undifferentiated pulp cells (Fig. 4d). The Notch activation, illustrated by EGFP expression, was strong in both regions (Fig. 4d).

**Loss of Notch causes MTACs premature differentiation**. To understand the function of the *Notch* pathway in the CL-MSCs and MTACs, we crossed *Collagen1α2 Cre* with *Rosa26R-LacZ* reporter mice[34] and found the LacZ staining specifically labeled some CL-MSCs cells but was absent from the incisor pulp and NVB-MSCs (Fig. 5a). We next investigated the consequence of losing *Notch* signaling on the CL-MSCs and subsequently in their descendants in the *Collagen1α2 Cre × RBP-Jkappa^flox/flox* mice. We performed immunostaining of DSP and DMP1 and found that unlike the wild-type (WT) mice, where DSP and DMP1 were mainly restrictively expressed in the odontoblast layer, in the *Collagen1α2 Cre × RBP-Jkappa^flox/flox* mice the two proteins' expression levels were highly elevated in the dental pulp cells (Fig. 5b). Particularly in those near odontoblast layers and DSP and DMP1 also started to be expressed by the MTACS (Fig. 5b). Hence losing *RBP-Jkappa* promoted early differentiation of MTACs. Consequently, we observed the mutant mice's dentin consisted of two layers of tissue, an outer layer with relative normal dentin structure and an inner layer with highly disorganized mineralized tissues, further confirming an enhanced

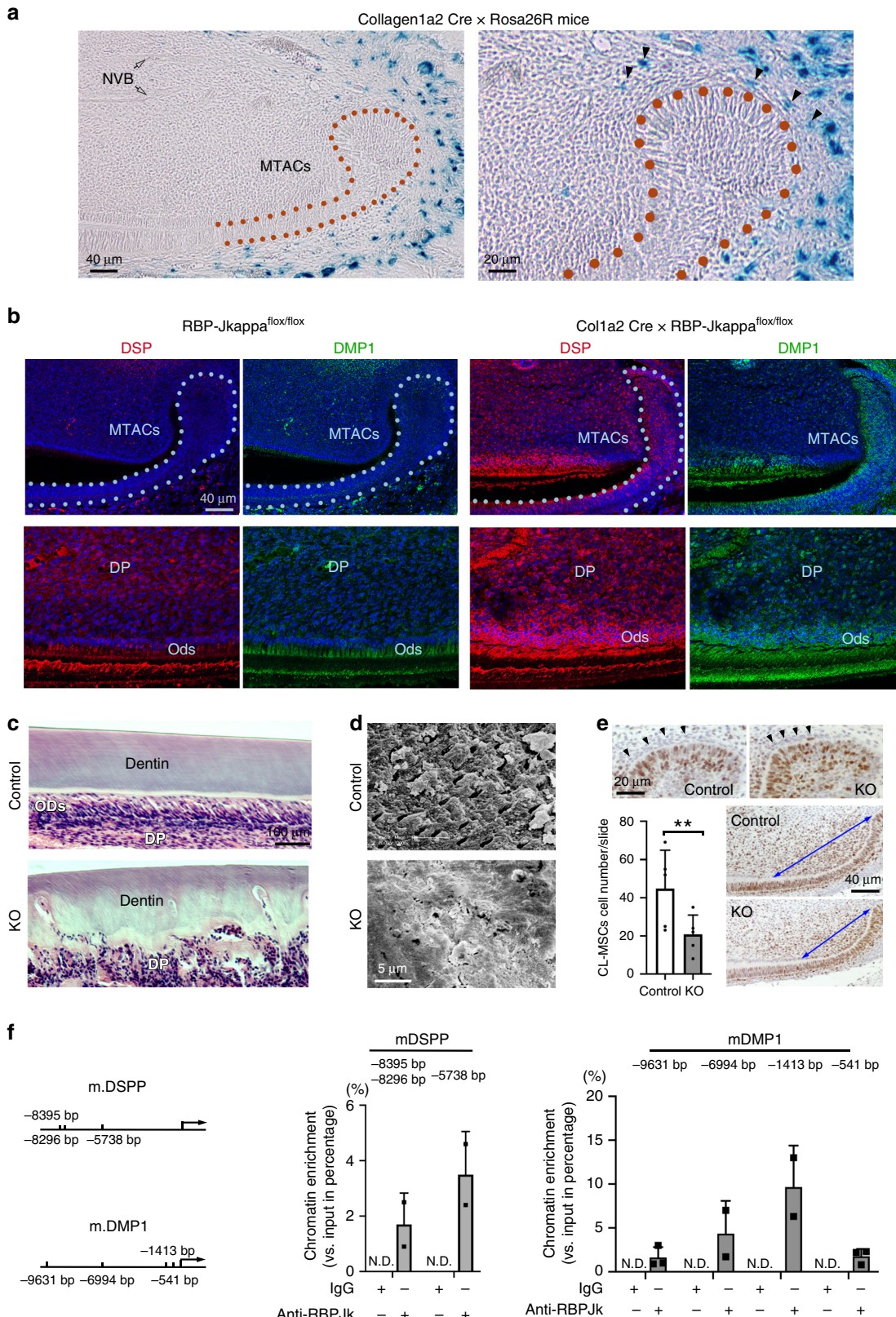

premature differentiation (Fig. 5c). Scanning electronic microscopy showed that in the mutant mice's incisor's second layer of dentin is a smear of disorganized dentin structure with ultra-structures entirely altered (Fig. 5d). Concurrently, the MTACs of the *Collagen1α2 Cre × RBP-Jkappa*<sup>flox/flox</sup> mice was significantly reduced as well as the cell number of CL-MSCs (Fig. 5e). The

finding that DSP and DMP1 were abnormally elevated in the absence of *RBP-Jkappa* in the MTACs and pulp cells suggested the possibility of *RBP-Jkappa* in performing a suppression role on the translation of *DSPP* and *DMP1* genes. We then investigated the promoters of the two genes in a tooth pulp-derived cell line, MO6-G3 cells[35], with chromatin immunoprecipitation (ChIP)

**Fig. 5** Loss of *Notch* pathway in CL-MSCs has impacts on MTACs and cell differentiation. **a** Representative LacZ staining on the P7 *Collagen1α2 Cre × Rosa26R* mouse lower incisor. Right panel represents the enlarged CL region of the left panel. Black arrows indicate the LacZ-positive cells in the CL-MSCs. Hollow arrows indicate the NVB region. **b** Immunofluorescence analysis of DSP and DMP1 expression in the *Collagen1a2 Cre × RBP-Jkappa flox/flox* mice incisor CL region vs. controls (*RBP-Jkappa flox/flox*). Nuclei were counterstained with DAPI. Light blue dotted lines indicate epithelial–mesenchymal junctions. **c** Hematoxylin–eosin staining of the *Collagen1a2 Cre × RBP-Jkappa flox/flox* (indicated as KO) mice incisor CL region vs. controls (*RBP-Jkappa flox/flox*, indicated as Control). Note the highly disturbed dentin formation. DP dental pulp, Ods odontoblasts. **d** Scanning electron microscopy micrographs on the 2-month-old Control and KO mouse incisor dentin. **e** Analysis of CL-MSCs and MTACs cell number in the Control and KO mice with Ki67 immunohistochemistry analysis (positive signal is visualized with brown DAB substrate). Two-way ANOVA followed by Bonferoni correction was performed. ** $p < 0.01$. Blue double-arrow-end lines outline the TAC regions contacting epithelium. **f** Chromatin IP analysis of RBP-Jkappa binding to the predicated sites (left) at mouse *DSPP* and *DMP1* gene promoters. The numbers represent the distance of the binding sites to the gene transcription starting site. The protein precipitation was performed using specific anti-RBP-Jkappa antibodies, with IgG as control. ND: not detected. Genomic DNA were extracted from $n = 5$ biologically independent animals and pooled. Triplicated samples were used for real time PCR analysis. Error bars represent standard deviation. Bars: **a** (left panel) and **b**: 40 μm; **a** (right panel) and **e**: 20 μm; **c**: 100 μm; **d**: 5 μm

and confirmed that indeed the mouse *DSPP* and *DMP1* promoters harbored functional RBP-Jkappa-binding sites (Fig. 5f). Therefore, these findings revealed that intercepting *Notch* signaling by deleting *RBP-Jkappa* in the CL-MSCs results in MTACs premature differentiation and abnormal dentin formation.

**Delta like 1 homolog (Dlk1) regulates MSC self-renewal**. To understand how the *Notch* signaling is modulated in different cellular compartments of the incisor mesenchyme, we next profiled Notch ligand expression by immunofluorescence and real-time RT-PCR analyses and found that Dlk1 was the only molecule that was exclusively expressed in the MTACs in the mesenchyme but not in the CL-MSCs (Fig. 6a–c). Dlk1 was also detected in tooth pulp cells and odontoblasts, but weaker by comparison to MTACs (Fig. 6a–c). We also noticed epithelial SCs and TACs also expressed Dlk1 but again at a lower level comparing to the MTACs (Fig. 6a). The findings suggested that *Dlk1* might have distinct roles in incisor MSC lineage patterning and differentiation. To determine whether *Dlk1* is a key molecule in the incisor MSCs, we analyzed *Dlk1 knockout (KO)* mice[36]. Using Ki67 as the MTACs marker and SmarcA2 as the CL-MSCs marker, we then observed significantly reduced MTACs and increased CL-MSCs numbers (Fig. 6d–f). Hence in the absence of *Dlk1*, CL-MSCs' activation was impeded. In the meantime, we also observed MTACs and tooth pulp cells starting expressing DSP and DMP1, suggesting a potential early differentiation (Fig. 6h), similar to the observation from the *Collagen 1 α2 Cre × RBP-Jkappa flox/flox* mice (Fig. 5b). Micro CT analysis further confirmed that the dentin or dentin like structures of the *Dlk1* KO mice was thicker than the WT mice (Fig. 6i, j). Therefore, losing *Dlk1* could mirror tooth defects of canonical *Notch* null mice.

As a potent Notch ligand, Dlk1 can act either as a cell membrane-bound form or as a secreted one. Given the lack of endogenous Dlk1expression in CL-MSCs, the latter can receive signals either as soluble form from MTACs and/or epithelial SCs, or as membrane-bound form through cell–cell contact with the neighboring epithelial SCs. To discern whether Dlk1 act on quiescent SC activation as a membrane bound or soluble form (or both), we grew MO6-G3 cells either on the Dlk1-coated dish or by adding soluble Dlk1 directly to the media. The effects of bound and free forms of Dlk1 differed: while bound form Dlk1 increased expression of the MSC markers (Fig. 7a), the free form increased those of MTAC marker expression (Fig. 7b). Also, soluble Dlk1 form enhanced cell growth enhanced cell growth (Fig. 7c).

To validate that *Dlk1* has the function to induce SCs to exit quiescence, we introduced a cell cycle indicator system: a modified Ki67p-T2A-FUCCI system (Fig. 7d)[37,38] into the MO6-G3 cells. The system allowed us to distinguish cell cycle phases being represented by different colors, and more importantly could facilitate the segregation of quiescent (G₀, colorless)

cells from those that have committed to cell cycle entry i.e. $G_1$/early S; mCherry-hCdt positive or late S/$G_2$/M mAG-hGem positive (Fig. 7e). With our system, we applied a classic cell cycle starvation and re-entry experiment. Under serum starvation for 4 days most cells entered $G_0$ with reduced numbers in $G_1$. The trend could be reversed by re-introducing serum (Fig. 7f). Therefore, with such setting, we observed adding Dlk1 onto the quiescent MO6-G3 cells could sufficiently trigger quiescent cells to enhance proliferation, i.e. to enter cell cycle.

To validate the significance of *Dlk1* in the incisor tooth, we adopted adult mouse incisor clipping model[23], which allowed us to trace the molecular changes of the stem cells upon stimulation. We first confirmed that in adult incisors, Notch1 and Notch2 expression remained the same as P7 (Supplementary Fig. 4a, Fig. 4c). After clipping, incisors grew faster than the control side (Supplementary Fig. 4b), with increased Dlk1 expression at the MTACs (Supplementary Fig. 4c). Together, using in vivo and in vitro approaches, we conclude *Dlk1* is a significant molecule in CL-MSCs maintenance and activation,

**Dlk1 CpG islands are dynamically methylated**. *Dlk1* is one of the known imprinted genes[39,40] that marks early mesenchymal precursors during embryonic development[41]. It also has a unique role in the other systems such as self-renewal status maintenance of MSCs[42,43]. How *Dlk1*'s activation and deactivation are controlled at epigenetic level has not been explored. *Dlk1* has two CpG islands, one at the 5′ beginning and the other at the 3′ end of the gene (Fig. 7g). We found in the incisor CL, two histone marks: H3k9me3 and H3k27me3 were both positive in CL-MSCs but highly reduced in the MTACs (Fig. 7h). We then explored *Dlk1* CpG island methylation status using MO6-G3 cells, and found that indeed the *Dlk1* gene's 5′ and 3′ CpG islands were highly methylated in starved condition and the methylations were abolished under growing condition (Fig. 7i). Hence the *Dlk1* gene activation is under dynamic epigenetic regulation when facing environmental changes.

**Dlk1 regulates MSC lineage differentiation**. The effective role of *Dlk1* in incisor MSCs activation and preservation prompted us to explore the possibility of translating the findings into potential regenerative medicine applications. The mouse and rat molar teeth are similar to human teeth in terms of development, biological structures, and pathological reactions. Unlike incisors, mouse molar tooth growth stops after the completion of root development and the tooth epithelium vanishes. However, the molar tooth's pulp, a mesenchymal tissue sharing similar embryonic origin as incisor mesenchyme, has an apical bud region that contains the MSCs that can contribute to limited regeneration of dentin upon injury or in caries[44]. We found the

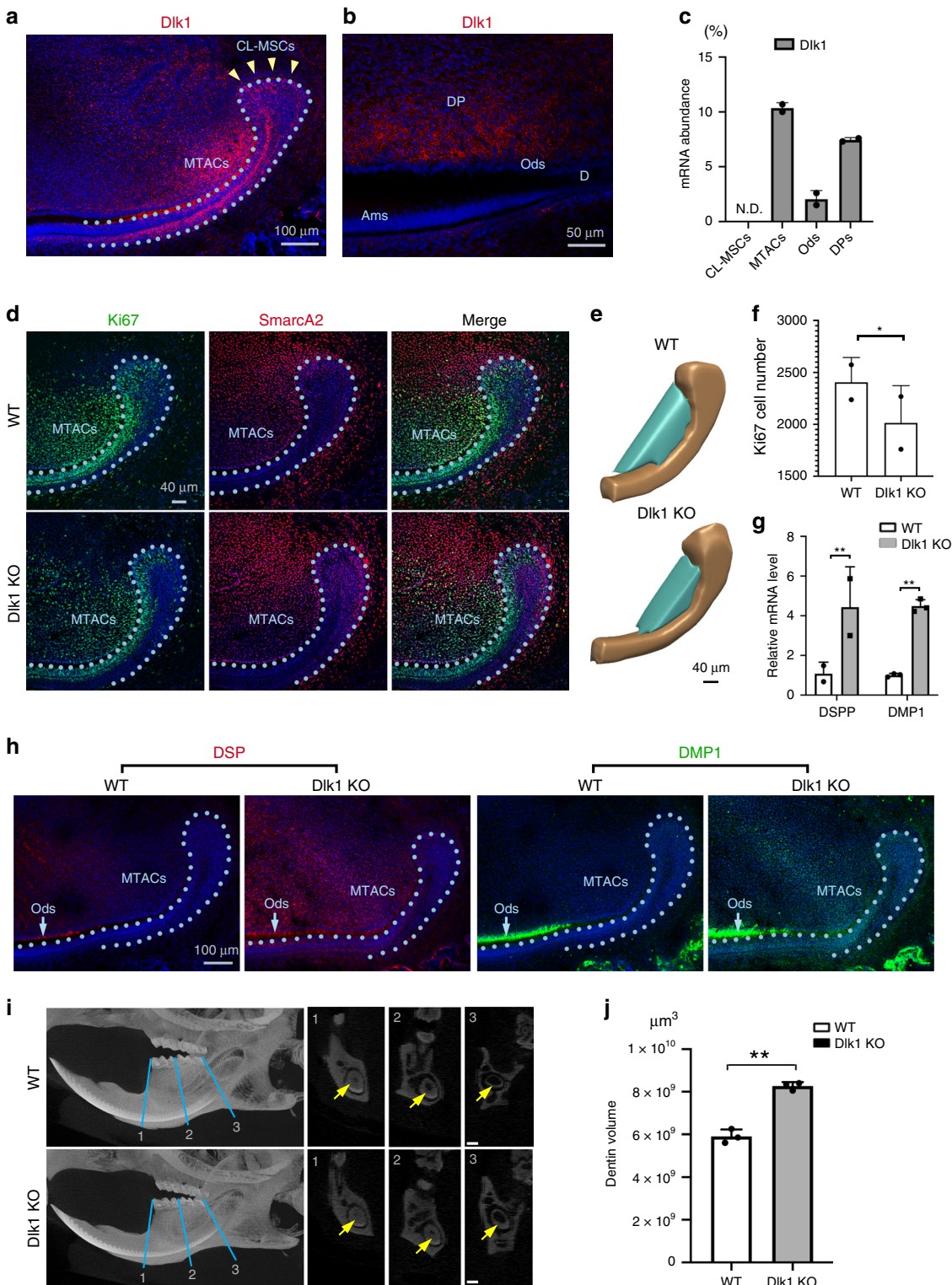

molar pulp apical bud cells also expressed *SmarcA2* and *Notch1* while the connecting differentiated odontoblasts expressed Notch2 and Dlk1 (Fig. 8a), suggesting that the molar MSCs might share with those of the incisor the same regulating molecular cascades. We therefore tested the capability of *Dlk1* in inducing MSC lineage differentiation in the molar teeth using tooth germ organ culture. Consistently, in the presence of Dlk1, we observed the treated tooth germs received enhanced dentin deposition (Fig. 8b). In the MO6-G3 cells, Dlk1 could also induce the mouse

*DSPP* gene's 1.5 kb promoter expression in a dose-dependent manner (Fig. 8c).

We next analyzed the effect of *Dlk1* overexpression on incisor and molar teeth using overexpressing *Dlk1* in mesenchymal cells under rat *Collagen I α1* promoter[43] (we named as *Col-Dlk1* Tg mice here). In adult mice, we observed the pulp of the *Col-Dlk1* Tg molar teeth were entirely filled with dentin like structures (Fig. 8d), while in the incisor tooth we observed the CL-MSCs marker: *SmarcA2* was significantly increased in the CL

**Fig. 6** Dlk1 prevents MTACs premature differentiation. **a, b** Immunofluorescence analysis of Dlk1 in the P7 incisor at CL region (**a**) and labial enamel–dentin differentiation region (**b**). Nuclei were counterstained with DAPI. Yellow arrowheads and light blue dotted line indicate CL-MSCs and epithelial–mesenchymal junction, respectively. Ams ameloblasts, d dentin, DP dental pulp, Ods odontoblasts. **c** Real-time RT-PCR analysis of Dlk1 in the indicated laser captured cell populations in the P7 CD1 mouse incisor from $n = 5$ biologically independent animals. Triplicated samples were used for real time RT-PCR analysis. ND: not detected. **d** Double immunofluorescence analysis of Ki67 and SmarcA2 in the wild type (WT) and Dlk1 KO mice at P7. Nuclei were counterstained with DAPI. Light blue dotted line indicates CL-MSCs and epithelial–mesenchymal junction. **e** 3D view of arbitrarily defined a countable MTACs region by aligning the anterior tip of the CL epithelium and first differentiated pre-odontoblasts (also see Fig. 5e). Ki67-positive cells in the cyan regions were quantified in **f**. **f** Quantification of Ki67-positive cell number at the cyan region indicated in **e**, from $n = 2$ biologically independent animals. Student's $t$-test was performed. *$p < 0.05$. **g, h** Real-time RT-PCR (**g**) and immunofluorescence analysis (**h**) of DSP and DMP1 in the P7 WT and Dlk1 KO mouse incisors. mRNA were extracted from $n = 5$ biologically independent animals. Two-way ANOVA followed by Bonferoni correction was performed. **$p < 0.01$. Light blue dotted line indicate CL-MSCs and epithelial–mesenchymal junction. **i** Micro-CT analysis of 2-month-old WT vs. Dlk1 KO mouse lower incisors. Yellow arrows indicate tooth pulp. Yellow arrows indicate dentin layer. **j** Dentin volume quantification of the 2-month-old WT ($n = 3$) vs. Dlk1 KO ($n = 3$) mouse lower incisors. Paired two-tailed Student's $t$-test was used for statistical analysis. **$p < 0.01$. Error bars represent standard deviation. Bars: **a** and **h**: 100 μm; **b**: 50 μm; **c**: 100 μm; **d** and **e**: 40 μm; **i**: 200 μm

mesenchymal cells. In fact, almost the entire MTAC region was filled with *SmarcA2* positive cells (Fig. 8e) while the Ki67-positive cells were highly reduced (comparing Fig. 8e with Fig. 6d). In the *Col-Dlk1* Tg mouse incisors, we also observed abnormal extra dentin formation (Fig. 8f, g). Together we proved *Dlk1* is indeed a very potent factor in promoting MSC lineage differentiation and SC preservation. However constitutively expressing *Dlk1* exhausted the MSCs pool at least in the molar teeth, but in the incisors *Dlk1* overexpression enhanced CL-MSCs presence in parallel to exhausting the TAC pool. The different consequence of *Dlk1* on incisor and molar teeth particularly in the *Col-Dlk1* Tg mice could be explained by the fact the epithelial–mesenchymal interactions at the incisor CL region could maintain incisor progenitor cells; however, in the fully developed molar where the tooth epithelium was absent, *Dlk1* functioned primarily as a trigger for the final differentiation of MSCs.

**Translational application**. We then strived to translate our findings in the mouse teeth into regenerative medicine. We found in the human teeth, under caries conditions (Fig. 9a), the site adjacent to the lesion showed that Notch2 ICD was initially expressed by the odontoblasts then the expression ceased (Fig. 9b). However, Dlk1 was always expressed by the odontoblasts or odontoblast like cells (Fig. 9b). Around the calcified pulp stone that represents abnormal tooth pulp cell differentiation and calcification, we also observed strong Notch2 ICD and Dlk1 expression (Fig. 9c). Hence the results suggest that *Dlk1* has a potentially key role in the tooth pulp repair and regeneration. In vitro, in two independent primary human tooth pulp cell lines, we observed that Dlk1 could significantly induce *DSPP* and *DMP1* mRNA expression in a dose-dependent manner (Fig. 9d). Finally, in an experimental tooth wound healing experiment (tooth pulp capping), where the rat molar pulp chamber was opened and covered either with a calcium hydroxide composition (Dycal®), a widely used clinical tooth pulp capping (covering the wound) reagent or Dycal plus Dlk1 (Fig. 9e), after 14 days, we observed the Dlk1-treated tooth pulp underwent significant reparative dentin formation compared to the control (Fig. 9f). In addition, the Dlk1-treated pulp expressed higher DSP and DMP1 (Fig. 9g). Therefore, *Dlk1* is able to enhance tooth pulp healing through enhancing MSCs lineage differentiation and proliferation.

## Discussion

A given SC niche has a specific location and composition within an individual tissue. It is generally accepted that solely one niche, or one kind of niche, exists in a tissue which can contribute to its homeostasis[3]. Recently, search of SCs in the context of tissue regeneration has challenged this concept and highlighted the

co-existence of distinct, and may be complementary niches within a tissue. The identification of intrinsic and extrinsic signals involved in SC activities could provide additional approaches with a therapeutic benefit. With this in mind, we report here that the mouse incisor tooth mesenchyme harbors a second niche named CL-MSC, besides the previously discovered NVB-MSC niche[21,22]. CL-MSC region has a unique anatomical organization with a diameter of approximately 50–100 μm, which surrounds the epithelial CL structure.

Why does a mouse incisor mesenchyme need two distinct MSC niches? Given that both of them have distinct anatomical locations: CL-MSC niche directly contacts epithelial SCs, while NVB-MSC niche is associated with nerve-blood vessel bundles[21], they might respond differently, and independently, to extracellular cues such as NVB and epithelial signals. Although the CL-MSC and NVB-MSC niches have no sharp physical boundary, they share nonetheless the expression of similar SC marker such as Thy-1 and PDGFrβ. They are both positive for *Gli1 Cre* and neighbor their associated tissues separately: the NVB and the incisor tooth epithelium, the tissues both produce SHH[17,22].

While NVB-MSCs mainly contribute to the replenishment of injured incisor pulp[21,23], CL-MSCs, on the other hand, are more responsible for giving raise to endogenous cell lineages such as MTACs and odontoblasts, which require a dynamic crosstalk with the tooth epithelial SCs, TACs, and lineage differentiated cells such as pre- and mature ameloblasts. It is possible that CL-MSCs and NVB-MSCs originate from the same progenitors during development, but acquire different lineage potentials once they arrive at the designated final anatomical destination. For example, both *Plp Cre* and *Pdgfrb Cre* can label neural crest cells[45] but *Pdgfrb Cre* appears can label broader scale of mesenchymal cells. This raises challenging questions such as do the two MSCs populations and/or niches have distinct and/or complementary roles in incisor development and wound healing? Do they share the same origin? From when and which stage do they start to acquire distinct phenotypes? Do MTAC population originate solely from CL-MSCs, or NVB-MSCs, or both? A detailed and specific lineage tracing analyses using *Cre* transgenics targeting individual MSCs populations might help to address these issues.

Unlike the incisor tooth, the molar tooth loses its epithelium after tooth eruption, and consequently, if any MSCs persist therein, those would be more like NVB-MSCs detected in the incisor tooth. A recent report have indeed suggested such a NVB-MSC niche in the molar teeth[46]. In addition, we report here that the apical papilla of the molar tooth is highly positive for *SmarcA2*, a DNA dependent ATPase. How the incisor and molar NVB-MSCs differ or are similar to each other and the linkage of NVB-MSCs with apical papilla in the molar teeth remain as two intriguing questions requiring further studies. We propose based

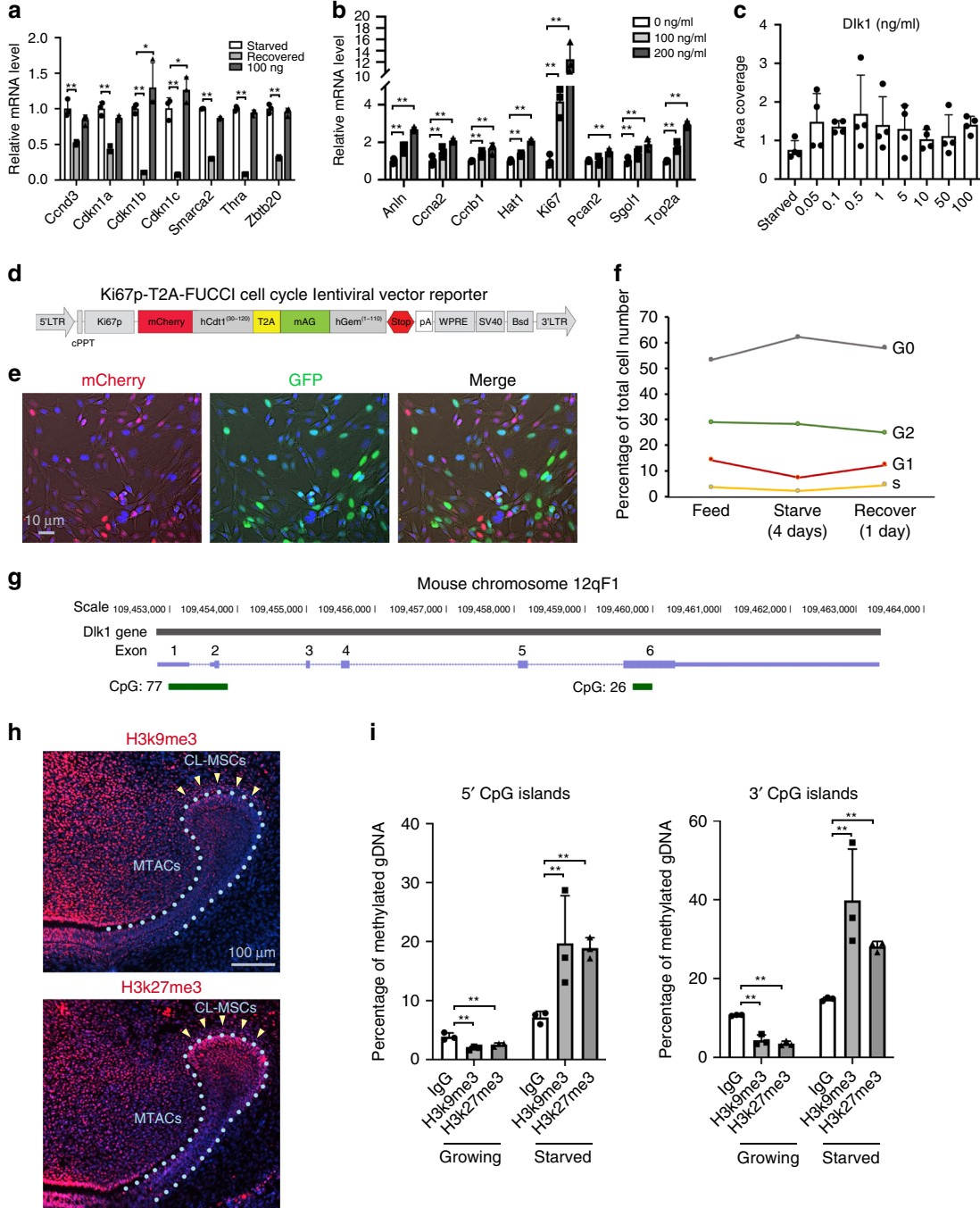

**Fig. 7** MTACs feedback to CL-MSCs through Dlk1. **a**, **b** Real time RT-PCR analysis of the indicated markers in the MO6-G3 cells cultured under bound (**a**, coating the protein onto cell culture dish first before seeding cells) and free form of Dlk1 (**b**, by adding directly on top of cultured cells). Cells were starved for nutrients for 4 days before recovery or Dlk1 treatment for 24 h as indicated. The results were from $n = 3$ biologically independent samples. Two-way ANOVA followed by Bonferoni correction was performed. No asterisk: $p > 0.05$; *$p < 0.05$; **$p < 0.01$. **c** Colony areas coverage of the soluble Dlk1-reated cells as indicated in **b**. The results were from $n = 3$ biologically independent samples. **d** Design of a new Ki67p-T2A based FUCCI cell cycle indicator. **e** Representative images of MO6-G3 cells incorporated with the Ki67p-T2A-FUCCI cells. Images were taken under phase contrast setting. **f** Flow cytometry analysis of MO6-G3/Ki67p-T2A-FUCCI cells at different cell cycle phase under indicated culture condition. **g** Illustration of genomic location of mouse *Dlk1* gene and its 5′ and 3′ CpG islands (green) and number. The positions of exons are indicated in purple. **h** Immunofluorescence analysis of H3k9me3 and H3k27me3 in the P7 mouse incisor. Nuclei were counterstained with DAPI. Yellow arrows mark clusters of odontoblasts that were EGFP positive. Light blue dotted lines indicate epithelial–mesenchymal junctions. **i** Methylation analysis of mouse *Dlk1* gene CpG islands in the MO6-G3 cells at growing and starved conditions. Genomic DNA were extracted from $n = 3$ biologically independent samples. Two-way ANOVA followed by Bonferoni correction was performed. **$p < 0.01$. Error bars represent standard deviation. Bars: **e**: 10 μm; **h**: 100 μm

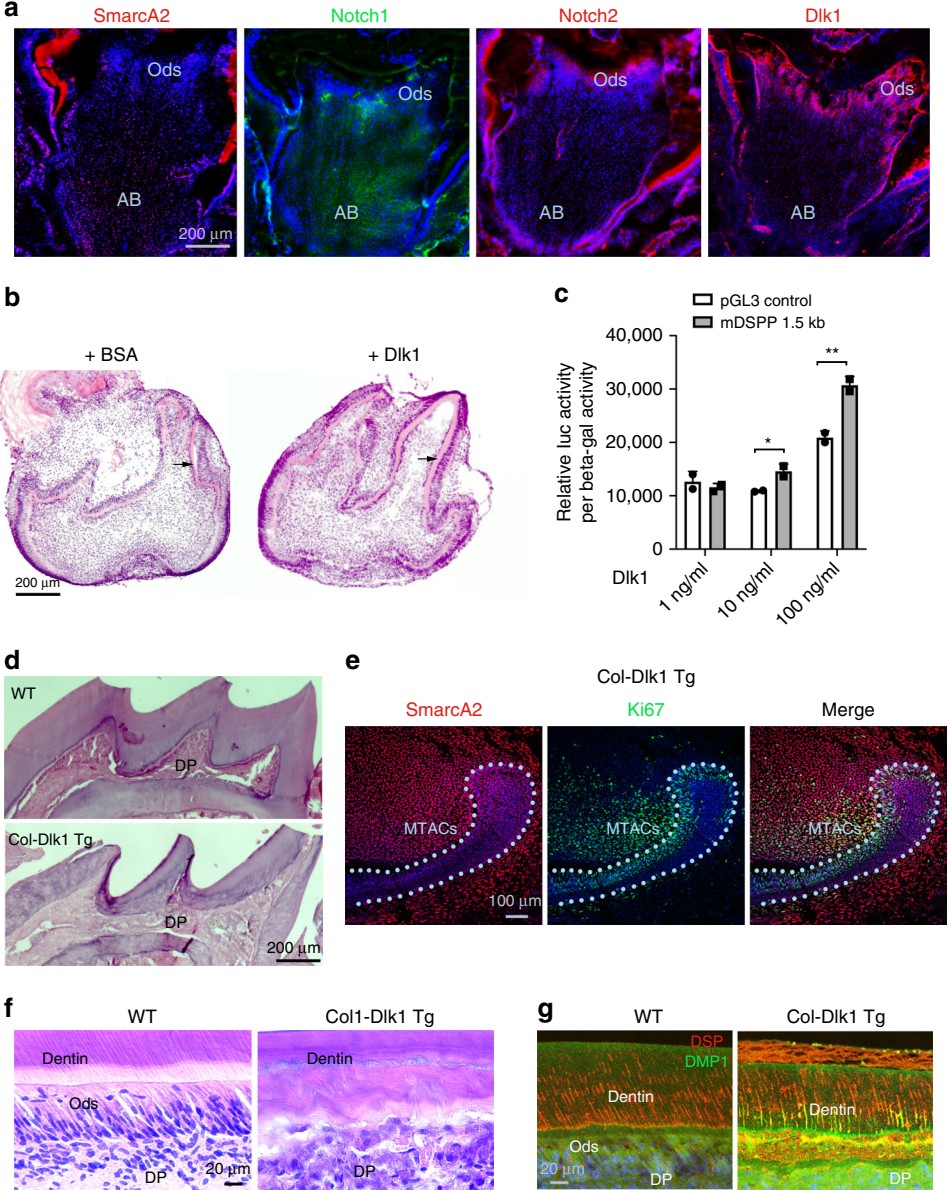

**Fig. 8** Dlk1 can be applied in enhancing MSCs lineage differentiation. **a** Immunofluorescence analysis of indicated molecules in the P7 CD1 mouse first lower molars. AB apical bud, Ods odontoblasts. **b** Representative images for organ cultures of E16.5 mouse first lower molar for 4 days in the absence or presence of Dlk1. Black arrows indicate dentin layers. Note the increased deposition of dentin in the Dlk1-treated samples. **c** Mouse DSPP promoter luciferase analysis in the MO6-G3 cells after receiving Dlk1 treatment at different concentrations. *$p < 0.05$; **$p < 0.01$. Genomic DNA were extracted from $n = 3$ biologically independent samples. Paired Student's t-test was performed. No asterisk: $p > 0.05$; *$p < 0.05$; **$p < 0.01$. **d** H&E images of the upper tooth crowns of the WT vs. *Col-Dlk1 Tg* mouse first lower molars. Note the mutant mouse's dental pulp (DP) is fully mineralized. **e** SmarcA2 and Ki67 double staining on *Col-Dlk1 Tg* mouse incisor at P7. Light blue dotted lines indicate epithelial–mesenchymal junctions. **f, g** H&E (**f**) and immunofluorescence analysis (**g**) of WT vs. *Col-Dlk1 Tg* mouse incisor labial dentin and tooth pulp. Error bars represent standard deviation. Bars: **a, b** and **d**: 10 μm; **e**: 100 μm; **f, g**: 20 μm

on the natures of the tooth MSCs that the NVB-associated pericytes or SCs are more responsible for stimulated replenishing following injury or wear, while the epithelial-associated MSCs such as CL-MSCs contribute to general tissue homeostasis.

In most tissues, TACs have limited proliferation capabilities and directly define the number of terminally differentiated cells. The number of the TACs needs to be calculated precisely by the tissue to ensure an accurate number of differentiating cells are generated upon request. Hence the signals that direct TAC proliferation requires precise fine tuning because insufficient activation of SCs results in developmental or tissue replenishment failure[47], while over activation such as forcing SCs to enter cell cycle can exhaust the SC pool[48]. In hair follicles, epithelial *SHH*

has a promoting role in SCs proliferation as well as in dermal signal regulation. We find that in incisor tooth, the mesenchymal *Dlk1* is uniquely expressed by the MTACs but absent in the CL-MSCs. *Dlk1* knockout mouse incisor has reduced CL-MSC number and premature differentiation, while *Dlk1* overexpression mice develop increased CL-MSC pool, providing a good example for how one molecule can balance the TACs with SCs, particularly in a mesenchymal tissue.

*Dlk1* an imprinted gene[40] that has been originally discovered in pre-adipocytes with a unique role in mediating MSC differentiation into various cell lineages, including osteoblast, adipocyte, and chondrocyte[42,43,49]. *Dlk1* has been shown either to activate[50] or inhibit *Notch* pathway[51,52], and the interaction

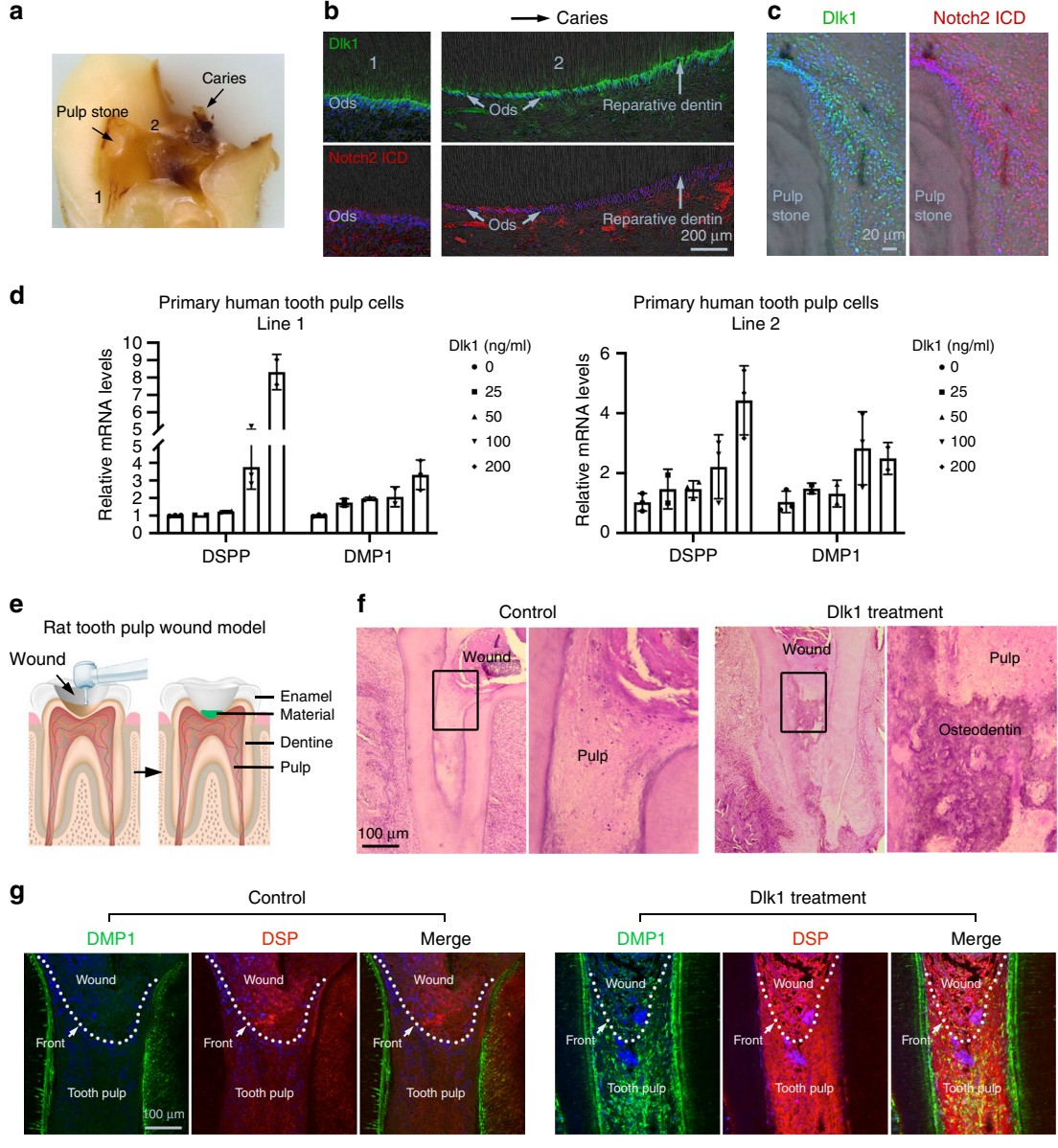

**Fig. 9** A translational strategy of applying Dlk1 in dental regenerative medicine. **a** Representative cross-sectional view of a human tooth with caries in paraffin- embedded block. Arrows indicate the caries site and a pulp stone in the tooth pulp. 1 indicates a relative healthy region and 2 indicates an adjacent region near the caries. **b**, **c** Immunofluorescent analysis of Dlk1 and Notch2 expression in the same sample of **a**, at different regions of the tooth. Pulp stone images were overlapped with phase contrast images to illustrate the stone location. **d** DSPP and DMP1 mRNA expression analysis with real-time RT-PCR in two independent human tooth pulp papilla cell lines treated with soluble Dlk1 at indicated concentrations (X axis). mRNA were extracted from $n = 3$ biologically independent samples. **e** Schematic illustration of the design of rat molar tooth capping model. **f**, **g** Representative H&E (**f**) and immunofluorescence (**g**) analyses of the rat molars received capping without (control) and with Dlk1 treatment for 7 days. Samples were immunolabelled for DMP1 and DSP and counterstained with DAPI. White dotted lines mark the wound fronts. Error bars represent standard deviation. Bars: **b**: 200 μm; **c**: 20 μm; **f**, **g**: 100 μm

between Dlk1 and Notch1 was demonstrated using two-hybrid system[50]. Here we shown that Dlk1 is co-expressed with Notch2 in the incisor MTACs, highlighting a potential existence of Dlk1-Notch2 regulation axis within MTAC population. The latter issue can be further dissected using approaches such as inducible *Cre* systems specifically targeting MTACs. Nonetheless, our study provides a functional insight into Dlk1's role in the maintenance of MTAC status as an inhibitor of precocious differentiation. In addition, Dlk1 has different roles in MSCs preservation and activation depending on if it reaches to MSCs through surface ligand-receptor bonding or as a free diffusible form. Hence a

given molecule could have a dual role on a SC population and its progeny.

It is important to note that in the murine incisor tooth the cellular complexity is beyond mesenchymal tissue. Both MTACs and CL-MSCs are in close vicinity to the epithelial compartment, where epithelial SCs and their progenies produced differentiated dental cells. Besides MTAC–CL-MSC communication, the mesenchymal–epithelial crosswalk can also be an essential feature to mediate the turnover of SCs and TACs associated with both cellular compartments. Our recent data have shown that, in addition to molecular signals such as SHH, the primary cilia

found in the epithelial compartment are able to sense such crosswalk[18].

In adult tissues, most of SCs are in a quiescent status and the majority their genes regulating their activation into TACs are silenced due to transcriptional suppression or epigenetic methylation[53]. The pulp of a molar tooth being a good example. Deciphering key molecules involved in such SC activation as well those implicated in SC niche maintenance can provide a unique molecular toolbox for regulating the tissue regeneration[1]. In the case of *Dlk1* gene, we uncover that its 5′ and 3′ CpG islands are dynamically methylated and demethylated in the absence of the nutrients (e.g., serum), suggesting that specific molecular targets controlling the SC activation are potential tools to stimulate regeneration processes. Further investigations to determine whether similar switch persists in other molecular pathways are urgently needed.

The regeneration outcomes differ in the mesenchymal compartment between incisor and molar. For an incisor, it regenerates normal dentin and enamel at its posterior part, i.e. the region connected to the CL, meanwhile its anterior pulp, which faces more external stimuli displays an irregular dentin formation. Previous lineage tracing experiments showed that the NVB-MSCs mainly contribute to the odontoblast formation at the anterior part of the tooth[21-23]. However, a mature molar tooth pulp, which solely contains NVB-MSC like cells, produces irregular osteodentin. Therefore, it is likely that NVB-MSCs are responsible for the rapid reactive repairing process, which is sensed and regulated by nerves to compensate the tissue loss. As an optimal medicine approach favors an entire structurally and functionally normal tissue regeneration, further dissection of the implication and regulation mechanisms of CL-MSCs vs. NVB-MSCs can provide deeper insights into how to translate developmental biology knowledge into regenerative medicine.

## Methods

**Animals**. All WT and transgenic animal breeding and operation procedures were approved by the institutional animal care and use committees at individual universities and in accordance with the guidelines and regulations for the care and use of laboratory animals, and compiled with all relevant ethical regulations for animal testing and research at each indicated countries and institutes, and received relevant ethical approvals. In details: CD1 mice at the University of Plymouth, UK and School of Stomatology, Capital Medical University, China; *Collagen1a2 Cre*, *RBP-Jkappa* [flox/flox], *Transgenic Notch Reporter* Mice at the University of Lausanne, Switzerland; *Dlk1*[−/−] mice at University of Castilla–La Mancha, Ciudad Real, Spain, *Collagen 1 α1 Dlk1 Tg* mice at the University of Southern Denmark, Denmark, *PDGFrβ Cre ER*[T2] and *ROSA*[mT/mG] at the Max Planck Institute for Molecular Biomedicine, Germany, and Wistar rats at Peking University, China.

**PDGFrβ mTG lineage tracing**. *PDGFrβ Cre ER*[T2] × *ROSA*[mT/mG] pups received intraperitoneal injections of 50 μg of tamoxifen (#T5648, Sigma) on postnatal days P0, P1 to induce Cre-mediated recombination. Samples were analyzed at P7 and P30. Tamoxifen stocks were prepared by dissolving 50 mg in 500 μl of ethanol and vortexing for 10 min before an equal volume of Kolliphor EL was added. One milligram aliquots were stored at −20 °C and dissolved in the required volume of phosphate-buffered saline (PBS) prior to injection. For analysis, P7 heads were frozen in liquid nitrogen, cryosectioned at 20 μm, and fixed with 4% paraformaldehyde (PFA) (#158127; Sigma Aldrich) solution in 10 mM PBS (#4417, Sigma Aldrich). Instead, P30 heads were fixed in 4% PFA first for overnight and washed in PBS and then passed 10%, 20%, and 30% sucrose in PBS solutions and cryosectioned directly without decalcification.

**Rat molar capping experiment**. Female Wistar rats at age 30 days were used. Bilateral wounds were created in the first molar to open up the pulp chamber with a 1 mm diameter dental drill. One wound was capped with a control hydroxyapatite substance (Dycal®), while the contralateral wound was capped with the same substance containing Dlk1 protein. Rats were sacrificed 14 days later. All the surgical operations were approved and monitored by the ethical committee of Peking University.

**Mouse incisor clipping experiment**. Female CD1 mice at age 8 weeks were used. Clipping of incisors was performed by removing one-third of the clinical crown on

the right lower incisor. Both incisors were notched on the surface above the gum line using a 1 mm diameter dental drill to monitor growth of the tooth. Mice were sacrificed 2 days after the procedure. All the surgical operations were approved and monitored by the ethical committee of Capital Medical University.

**Human tooth samples**. The third molars extracted from patients with Class I caries due to orthodontic needs but without pulpitis symptoms were collected at the Tian Jin Medical University, Hospital of Stomatology, with the consent of the patients, under the direction and with the approval of the university ethical committee. The samples were fixed in 10% neutralized formalin (pH 7) for 7 days followed by decalcification in 30% EDTA (EDTA, #BP2482-100; Thermo Fisher) for 90 days before embedding with paraffin.

**Animal tissues preparation**. For P7 mice, tissues were either immediately submerged in dry ice cooled 2-methylbutane (#M32631; Sigma Aldrich) for 10 min or directly frozen in liquid nitrogen for 5 min. Frozen tissues were then stored at −80 °C. For adult mice, mandibles were dissected and fixed in 10% formalin (#HT50-128; Sigma Aldrich) for 24 h before being decalcified in 14% EDTA for at least 30 days before embedding.

**Dentin slice preparation**. Incisors of P7 or P9 CD1 mice were extracted by microdissection under a Leica M80 stereo microscope. The dentin of the incisor was isolated, washed, and stored in Hank's balanced salt solution (HBSS, #14175-053, Gibco) for 24 h. The incisors were extracted and the CL region isolated. The tissue was placed into 1% Dispase II (#4942078001; Roche), which was dissolved in HBSS containing 1% penicillin–streptomycin (#SV30079.01; Hyclone) and filtered using a 0.22-μm filter, for 1 h at 37 °C. The tissues then were washed with 10 ml of Dulbecco's modified Eagle's medium (DMEM; #31966021; Gibco) containing 10% fetal bovine serum (FBS, #F7524; Sigma Aldrich) and 1% penicillin–streptomycin (#SV30079.01; Hyclone). The epithelial and mesenchymal tissues were then mechanically dissociated using 23G needles under a Leica M80 stereo microscope.

The dentin slices were placed on sterile cell strainers (#431750; Corning Inc) into a six-well plate (#140685; Fisher Scientific) containing DMEM/F12 (DMEM-F12, #31331-028; Gibco) with 20% FBS, 1% penicillin–streptomycin, and 1% L-ascorbic acid (w/w) (#A4403; Sigma Aldrich). The extracted incisor CL-MSC mesenchyme was then placed on top of the dentin and cultured for 4–7 days. The dentin–mesenchyme samples were removed from culture and placed directly into 4% PFA (#158127; Sigma Aldrich) solution in 10 mM PBS (#4417; Sigma Aldrich) for 20 min at room temperature. After fixation, the samples were washed twice in PBS. Samples were then passed through serial solutions of 10%, 20%, and 30% sucrose (#S0389; Sigma Aldrich) in PBS for 20 min each, before being embedded in Tissue-Tek® O.C.T.™ (#4583; Sakura) and snap frozen by plunging into liquid nitrogen. Once frozen samples were stored at −80 °C before being sectioned at 15 μm and stained for immunofluorescent analysis. Alternatively, sections were used to visualize the actin cytoskeleton. Samples were cryosectioned at 15 μm thickness on a Leica CM1850 cryostat onto Polysine™ Microscope Adhesion Slides (#J2800AMNZ; Thermo Scientific).

**Semi-solid organ culture**. Semi-solid culture conditions were set up using culture media supplemented with agar[54]. Briefly, prewarmed DMEM-F12 supplemented with 20% FBS and 1% penicillin–streptomycin (complete medium) was mixed 1:10 with 2.5% (w/w) agar (#A1296; Sigma) dissolved in sterile water. Two milliliters of media/agar mix was added to each well of a six-well-plate and allowed to set at room temperature for 30 min, before tissue was placed on top and the dish was covered with complete medium.

**Immunofluorescence staining**. For detail antibody information, please see Supplementary Table 1. Preparation of formalin-fixed paraffin-embedded (FFPE) samples is performed as described here; FFPE samples were sectioned at a thickness of 10 μm, and placed onto Superfrost slides (#Z692255; Sigma Aldrich). After drying overnight, deparaffinization was performed. Slides were heated to 55 °C for 20 min before being twice washed in xylenes (#534056; Sigma Aldrich) for 10 min. Slides were then washed in 100% industrial methylated spirits (IMS) (#23684.360, VWR) for 5 min, before being washed for 2 min in 95% IMS and then 70% IMS. Antigen retrieval was performed by microwaving the slides in a 0.01 M citrate buffer solution (citric acid and 0.05% Tween-20, #C2404 and P9416, respectively; Sigma Aldrich) for 1 min.

Once prepared all sample types were washed three times in PBST (PBS containing 0.1% Triton X-100 (Sigma Aldrich)) for 5 min per wash. Non-specific binding was blocked by incubation with PBST containing 5% Donkey Serum (#D9663; Sigma Aldrich), 0.25% cold water fish gelatine (#G7765; Sigma Aldrich), and 0.25% albumin bovine serum (BSA; #A2153; Sigma Aldrich) for 60 min. Primary antibodies were incubated overnight at 4 °C. Slides were washed thrice in PBST before incubation with secondary antibodies for 2 h at room temperature. Nuclei were counterstained with 4′,6-diamidino-2-phenylindole (DAPI, 2 μg/ml, #D9542; Sigma Aldrich) for 10 min.

Immunofluorescence images were captured using a Leica DMI6000 confocal microscope with a Leica TCS SP8 attachment at a scanning thickness of 1 μm per section. The microscope was running LAS AF software from Leica. Post

imaging assembling and processing was conducted using Adobe Photoshop CC. Preparation and fixation of cells to be used for immunofluorescence (IF) analysis was performed. Cells were washed in HBSS, and then fixed in ice cooled 4% PFA solution in 10 mM PBS for 30 min. Preparation and fixation of frozen tissue is as follows; frozen tissue was cryosectioned at 15-μm thickness on a Leica CM1850 cryostat. Sections were mounted onto Polysine™ Microscope Adhesion Slides and allowed to air dry for 30 min before fixation in ice cold acetone (#34850; Sigma Aldrich) or freshly made ice cooled 4% PFA solution for 30 min.

Apoptotic cell death was analyzed using Fluorescein In Situ Cell Death Detection Kit (#11684795910, Roche) according to the manufacturer's instructions. Briefly, samples were fixed in ice cold 4% PFA at RT for 30 min and washed twice in PBS. TUNEL reaction mixture was prepared by mixing 50 μl Enzyme solution to 450 μl Label solution, 50 μl of TUNEL reaction mixture was added to sample and incubated in a humidified atmosphere for 3 h at 37 °C in the dark. Slides were then washed three times with PBS before mounting with DAKO fluorescence mounting medium (#S3023; Aligent). Images were captured using a Leica DMI6000 confocal microscope with a Leica TCS SP8 attachment. The microscope ran LAS AF software from Leica (3.5.2.18963).

**Hematoxylin & eosin staining.** Slides of FFPE-sectioned tissue are stained using hematoxylin and eosin histological dyes according to the following protocol. FFPE samples were sectioned at a thickness of 10 μm, and placed onto Superfrost® slides. After drying overnight, deparaffinization was performed. Slides were heated to 55 °C for 20 min. Subsequently slides were washed twice in xylenes for 5 min. The tissue is then rehydrated through two changes of 100% IMS, followed by a wash in 95% IMS and then 70% IMS, each for 2 min. The slides were briefly washed in distilled water before being stained for 8 min in Harris hematoxylin (#HHS16; Sigma). After staining was sufficient the slides were washed in tap water for 5 min. Differentiation of the stain was achieved by placing the slides in 1% acid alcohol (#56694; Sigma Aldrich) for 30 s, before a further 5 min tap water wash. Subsequent counterstaining was performed by washing the slides briefly in 95% IMS, followed by a 1-min incubation in 0.25% eosin Y solution (#230251; Sigma). After staining was complete, the tissue was dehydrated by passing the slides through two 5 min washes in 95% IMS and then 100% IMS. In order to clear the sample, the slides were washed twice more in xylenes for 5 min before being mounted using Eukitt, xylene-based mounting medium (#03989; Fluka).

**Cell culture.** Established mouse molar mesenchyme cell line MO6-G3 cells[35] were cultured in DMEM containing 10% FBS, 1% penicillin–streptomycin, and 300 μg/ml Geneticin (#10131019; Gibco). Cervical loop mesenchyme cells were isolated as follows: incisors were dissected from postnatal day 30 CD1 mice shortly after death. The CL region was dissected under a Leica M80 stereomicroscope. The explants were incubated for 60 min in 1% type I collagenase (#C0130; Sigma) in HBSS with 1% penicillin–streptomycin and 1% Fungizone® Antimycotic (#15290-018; Gibco). Following incubation collagenase was neutralized with DMEM/F12 containing 10% FBS and 1% penicillin–streptomycin. The resulting cell suspension was centrifuged at 1000 r.c.f., at room temperature for 5 min. Supernatant was discarded and the cell pellet was resuspended in complete AmnioMax Media, which combines AmnioMAX™ C-100 Basal Medium (#17001-07; Gibco) and AmnioMAX™ C-100 Supplement (#12556-023; Gibco) before being cultured on poly-D-lysine CELLCOAT® dishes (#628940; Greiner Bio One). Primary human tooth pulp cell lines were collected from young permanent teeth that were extracted due to orthodontic treatment requirement[55], with the consent of the patients, under the direction and with the approval of the Ethical Committee of Beijing University. The cells were maintained in DMEM-F12 (#31331-028; Gibco) containing 20% FBS and 1% penicillin–streptomycin.

**Treatment of cultured cells with recombinant proteins.** Cells were treated with recombinant proteins either through direct addition of the protein to fresh media or via an indirect coating method. Direct protein treatment involved the addition of the protein into fresh un-supplemented or fully supplemented media as appropriate to the cell line. This fresh media was then placed onto the cells that had been cultured for 24 h. Media containing the protein was replaced every 48 h. The indirect binding method involved passaging cells onto pre-prepared dishes that had already been coated with protein. Dish preparation was performed as follows: six-well plates were coated by placing 10 μg/ml of goat anti-human IgG (#I1886; Sigma) or Goat anti-Mouse IgG (#M8642; Sigma) in HBSS into the dish. The dish was then incubated for 30 min at 37 °C. Dishes were quickly washed five times in HBSS and then blocked by incubating with a 2% solution of BSA in HBSS overnight at 4 °C. Following blocking 10 μg/ml of the recombinant protein is added to the blocking solution, and incubated for 2 h at 37 °C. Following preparation of the dishes they are washed quickly in HBSS five times and immediately cells are plated into them.

**MSCs multipotential differentiation assays.** C57 Black 6 mice sacrificed at P7. CL dissected out and digested in 1% collagenase (in PBS; #C0130; Sigma), 1% AA (#11570486; Gibco), 1% Amphotericin B (#15290018; Gibco) in HBSS for 1 h at 37 °C with frequent agitation. Following incubation, collagenase was neutralized with DMEM + 10% FBS was added to the cell suspension for 10 min, and cells

were subsequently spun at 1 r.c.f. for 5 min. The supernatant was discarded and the pellet re-suspended in AmnioMax basal media, 14% AmnioMax supplement, 0.1% ascorbic acid, and 1% AA. CL-MSC cells were amplified for 2 days in vitro before seeding into 96-well plates at a seeding density of $1 \times 10^4$ cells per well for osteogenic/adipogenic differentiation. Cell culture media was changed to StemX-Vivo osteogenic/adipogenic base media (#CCM007; R&D Biosystems) with 1% AA. Following 48 h of culture, osteogenic or adipogenic cell differentiation was induced by supplementing the base media with 5% StemXVivo osteogenic supplement (#CCM008/CM009; R&D Systems) or 1% StemXVivo adipogenic supplement (#CCM011; R&D Systems) respectively. Media was replaced every 48 h. Following 21 days of culture cells were fixed in 4% PFA at RT for 30 min and washed twice in PBS ready for staining.

For chondrogenic differentiation $2 \times 10^6$ CL-MSCs were pelleted by centrifugation at 0.2 r.c.f. for 5 min. Media was replaced with 1 ml StemXVivo chondrogenic base media (#CCM005; R&D Systems), 1% StemXVivo chondrogenic supplement (#CCM006; R&D Systems), and 1% AA. Media was removed and replaced with complete Chondrogenic differentiation media every 2–3 days. Chondrogenic pellets were harvested at 15 days for analysis. Pellets were fixed in 4% PFA for 1 h at RT, followed by incubation in sucrose gradients before embedding in OCT and freezing using LN for sectioning.

Adipogenic differentiation was analyzed using 4,4-difluoro-1,3,5,7,8-pentamethyl-4-Bora-3a,4a-Diaza-s-Indacene (BODIPY 493/503; #D3922; Invitrogen). In all, 2 μg/ml 4′-6-diamidino-2-phenylindole in 10 mM PBS (1:10,000; DAPI; #D9542; Sigma-Aldrich) was mixed with 0.5 nM BODIPY for 20 min on a rocker at RT. Cells washed 10 times with PBS before mounting with DAKO fluorescence mounting medium (#S3023; Aligent). Images were captured using a Leica DMI6000 confocal microscope with a Leica TCS SP8 attachment. The microscope ran LAS AF software from Leica (3.5.2.18963).

Osteogenic differentiation was analyzed using a 2% Alizarin Red solution (#A5533 25 G; Sigma), prepared by dissolving 0.2 g in 10 ml water, filtering, and then adjusting to pH 4.2 by addition of NaOH/HCl. Fixed cells were washed in dH₂O and then stained at room temperature for 45 min in the dark. Excess solution was removed by washing with dH₂O. Images were taken using a Leica M80 stereomicroscope and a Leica MC170 HD camera.

Frozen chondrocyte pellets were sectioned at 20 μm on polysine slides and air dried. OCT was dissolved by washing in PBS and then pellets were washed in 3% acetic acid for 3 min. Staining was achieved with a 1% Alcian Blue solution (#TMS-010-C; Millipore) at room temperature for 30 min. A further wash in 3% acetic acid (Sigma) was used to remove excess staining. Nuclei were counterstained with Nuclear Fast Red solution at room temperature for 5 min. Sections were dehydrated using an ethanol gradient, cleared in xylene (534056, Honeywell Research Chemicals), and mounted with Eukitt quick hardening mounting media (#03989; Fluka).

**Generation of Ki67p-T2A-Fucci cell cycle reporter.** The FUCCI plasmids mAG-hGeminin(1/110)-pCSII-EF-MCS and mCherry-hCdt1(30/120)-pCSII-EF-MCS were kindly provided by Dr. Atsushi Miyawaki[37]. Open reading frames for mAG-hGeminin (1/110) and mCherry-hCdt1 (30/120) were PCR sub-cloned into pENTR-D-Topo plasmids (Life Technologies). Gibson Assembly (NEBuilder HiFi DNA Assembly Kit, New England Biolabs) was used to fuse the mAG-hGeminin (1/110) to the C-terminus of mCherry-hCdt1 (30/120) separated by a ribosomal skip T2A sequence (T2A-FUCCI) according to the manufacturer's instructions. The coding region was verified by direct sequencing. Gateway recombination was then used to generate a lentiviral vector containing the 1.5 kb human Ki67 proximal-promoter (Ki67p)[38] upstream of T2A-FUCCI open reading frame in the 2k7Bsd lentiviral vector[56]. MO6-G3 were incubated with the lentiviral supernatant, obtained as described in "Generation of Ki67p-T2A-Fucci cell cycle reporter", and 10 μg/ml polybrene (Merck) overnight under normal culture conditions. After 2 h, the viral supernatant was replaced with normal growth medium. Infected MO6-G3 cells were selected indefinitely using 10 μg/ml blasticidin (Sigma-Aldrich)[18].

**Synchronization of cell cycle.** Twenty-four hours after initial seeding and culture under normal cell culture conditions (see above), cells were exposed to basal or fully supplemented media. Four days later, they were replenished with new media, either maintaining the current supplement with or without supplements.

**Cell culture confluency assay.** Cells were seeded at equal densities into a six-well plate. In certain conditions siRNA was added to the culture at the point of replenishing the supplements to the media. Cells were fixed in 10% formalin for 30 min at room temperature before being washed three times for 5 min each in HBSS. Cells were visualized using 0.01% crystal violet solution (#V5265; Sigma). Images were taken of four distinct areas of each well using a Leica DM1000 LED microscope. Cell confluency was measured using Fiji (Image J 1.51n).

**Flow cytometric and FACS analysis of cultured cells.** Ki67-FUCCI system infected MO6-G3 cells were harvested, fixed in 4% PFA in 10 mM PBS for 30 min at room temperature, and analyzed by flow cytometry using the BD FACSCanto II SOR (Beckman Coulter). Data were acquired using blue laser (488 nm) for

mAzamiGreen signal and yellow-green laser (561 nm) for mCherry and analyzed using the FlowJo software v10.3 (Tree Star Inc.).

Cultured cells were harvested and fixed using flow staining buffer (#FC001, R&D) for 15 min at room temperature. They were then immunolabelled with a panel of antibodies provided from the Mouse Mesenchymal Stem Cell Marker Antibody Panel kit (#SC018, R&D) for 30 min at room temperature. Afterward, they were washed and incubated with donkey secondary antibodies Alexa 488 anti-rat IgG (#A21208; Life Technologies) for 30 min. Labeled cells were observed using a BD Accuri C6 (Beckman Coulter) and analysis with BD Accuri C6 Software (Accuri Cytometers Inc., version 1.0.264.21).

**Chromatin immunoprecipitation**. ChIP was performed on cells using the ChIP-IT high sensitivity kit (#53040; Active Motif) in accordance with the provided protocol. In summary the procedure was as follows; cells were fixed using Complete Cell Fixation Solution added directly to the culture media for 15 min at room temperature. Fixation was quenched using Stop Solution. Cells were scraped and collected by centrifugation at 1250 r.c.f. for 3 min at 4 °C. The pellet was resuspended and washed twice in ice-cold PBS. The resultant pellet was resuspended in Chromatin Prep Buffer supplemented with protease inhibitor cocktail (PIC) and phenylmethylsulfonyl fluoride (PMSF) and incubated for 10 min on ice. Using a Dounce homogenizer the cells were homogenized, collected by centrifugation, and resuspended in fresh ChIP Buffer supplemented with PIC and PMSF. Chromatin was sheared using a Diagenode Bioruptor Pico (Diagenode) by sonicating for 15 s followed by 30 s of rest for four cycles. Cell debris was removed by pelleting this out by centrifugation. Antibodies or control IgG was mixed with Blocker as outlined in Supplementary Table 1; this antibody mix was mixed with ChIP Buffer, PIC, and sonicated chromatin and incubated overnight at 4 °C. Immunoprecipitation was performed by washing Protein G agarose beads in TE pH 8.8 and then mixing the washed beads to the antibody/chromatin mixture and incubating for 3 h at 4 °C. Following incubation with the beads the solution was passed through a ChIP filtration column to collect the bound chromatin. The bound chromatin was washed and eluted using the provided solutions. The cross-links within the eluted sample was reversed and the DNA purified by overnight incubation with Proteinase K at 37 °C. The DNA was collected by heating the sample to 95 °C for 8 min and then plunging into ice for 1 min to deactivate the proteinase K function. DNA Purification Binding Buffer and sodium acetate was added to the sample, which was then passed through a DNA Purification Column by centrifugation. The latter was washed with DNA Purification Wash Buffer, and DNA was eluted using DNA Purification Elution Buffer. Input DNA was prepared in the same manner, however, rather than column-based purification. Following proteinase K treatment, DNA was immunoprecipitated in 2-propanol (#34965; Fluka Analytical) with 1 μl GlycoBlue (#AM9515; Ambion), before being washed in 70% ethanol (#20821.321; VWR Chemicals) and resuspended in 0.1% diethylpyrocarbonate (#D5758; Sigma Aldrich)-treated distilled water (DEPC).

**LCM of frozen tissues**. Frozen tissue was sectioned using a Leica CM1850 cryostat at a thickness of 20 μm. Sections were mounted onto PEN Membrane Glass Slides (#LCM0522; Applied Biosystems). FFPE blocks were sectioned on a Microm HM320 microtome at a thickness of 10 μm, and placed onto PEN membrane glass slides. All prepared slides were stained using 1% methyl green (#67060; Fluka Analytical) in 0.1% DEPC-treated distilled water, then washed three times for 30 s in 0.1% DEPC-treated distilled water before being allowed to dry for 5 min. An ArcturusXT™ LCM instrument was used to perform LCM onto CapSure® Macro LCM Caps (#LCM0211; Applied Biosystems). Captured tissue to be used for RNA analysis was stored in Tri-Reagent (#T9424; Sigma Aldrich) at −80 °C within 30 min of initial sectioning. Captured tissue to be used for gDNA analysis was processed immediately.

**mRNA extraction from cultured cells and LCM captured tissue**. Cells in cultured were washed with HBSS, incubated in Tri-Reagent for 5 min at room temperature, and then scrapped. Cell suspension (500 μl) was collected. Alternatively, tissue samples were collected using LCM, and their RNA was extracted and purified using an acid guanidinium thiocyanate–phenol–chloroform extraction protocol. Briefly, total RNA was extracted using Tri-Reagent and the samples were vortexed, incubated for 5 min at room temperature before the addition of chloroform (#C2432; Sigma Aldrich) at a ratio of 5:1 (lysate: chloroform). Samples were incubated for another 5 min and then centrifugated at 13,000 r.p.m. for 10 min. The aqueous phase was collected, and added 1:1 with 2-propanol with 1 μl GlycoBlue. Samples were incubated at −20 °C overnight, and afterward centrifugated at 13,000 r.p.m. for 45 min at 4 °C. The subsequent pellets were washed in 1 ml 70% ethanol and centrifuged at 9000 r.p.m. for 10 min at room temperature. Finally, pellets were resuspended in 10 μl of 0.1% DEPC-treated distilled water. Purified RNA was quantified on a NanoDrop 2000 UV-Vis Spectrophotometer (Thermo Scientific). Quality control was assessed by analysis of the 260/230 ratio, which highlights phenol contamination, and 260/280 ratio more than 1.8 which can be used as an indicator of DNA contamination.

Reverse transcription was performed with High-Capacity cDNA Reverse Transcription Kit (#4368814; Applied Biosystems) in accordance with the

manufacturer's protocol using a Veriti™ Thermal Cycler 96 well (Applied Biosystems). The following program was applied: 25 °C for 10 min, 37 °C for 120 min, 28 °C for 5 min and 4 °C indefinitely thereafter. Samples were diluted in 180 μl of 0.1% DEPC-treated distilled water and stored at −20 °C.

**Real-time PCR and data analysis**. Real-time RT-PCR analysis was performed on a LightCycler 480 Real-Time system (Roche) for 45 cycles, using a SYBR Green I MasterMix (#4887352001; Roche Life Science). Internal control primers were used to determine relative quantification of gene expression using the $2^{-\Delta\Delta Ct}$ method. Analyses were performed using three technical replicates. Experimental data were analyzed using PRISM 5 software (Graph Pad Software). For Real-Time PCR analysis, a two-way ANOVA followed by Bonferoni correction was performed. Statistical significance was set at *$p < 0.05$, **$p < 0.01$. For individual primer sequence please see Supplementary Table 1.

**Three-dimensional reconstruction**. For dentin culture assay, immuno-fluorescence micrographes were captured using A Leica DMI6000 confocal microscope with a Leica TCS SP8 attachment at a scanning thickness of 1 μm per section. Microscope was running LAS AF software from Leica. These images were used to reconstruct the growth of the cells into the dentine tubules using Imaris (version 9.0.2) (Bitplane). The three-dimensional (3D) reconstruction of the CL and surrounding transit amplifying mesenchyme region was performed by taking serial immunofluorescent-labeled cryosections. The epithelium was outlined from approximately 20 consecutive 20-μm-thick sagittal sections of the apical end of the mouse incisor.

The corresponding TAC region was defined by drawing a straight line between the epithelial tip, where marks boundary the epithelial stem cells and TACs on serial sections containing CL. 3D reconstruction of the region was performed with computational reconstruction using BioVis software (version: 3.1.1.11) (http://www.biovis3d.com). The Ki67-positive cells were then quantified on each section and total cell number were counted based on DAPI staining.

**Micro-computed tomography (micro-CT)**. Whole animal heads were scanned with a SkyScan-1072 desktop micro-CT system at a resolution of 9 μm. The generated data were then analyzed with Imaris software (version 9.0.2, Bitplane).

**Scanning microscopy**. Isolated incisor teeth were cut transversally at the tooth–bone junction with a No. 11 scalpel then fixed in 3% glutaraldehyde (Sigma-Aldrich) in PBS for 4 h then incubated in ethanol for 5 min each at a concentration of 50%, 70%, 90%, and 95%, and three times for 30 min 100%. Samples were then dried for 24 h, gold sputtered, and observed under a XL20 scanning microscope (Philips).

**DSPP promoter**. The mouse *DSPP* promoter plasmids[57] were heat-shock treated then introduced into *E. coli* DH5α and grown overnight at appropriate conditions and under ampicillin selection. Plasmids were purified using the Plasmid Maxi Kit (#12163; Qiagen) according to the manufacturer's recommendations. The *DSPP* reporter plasmids and empty pGL-3 basic luciferase vector were then introduced into M06-G3 using JetPEI (#101-10N; Polyplus).

**X-gal staining**. The X-gal staining on the LacZ reporter mouse incisors was performed either with whole mount staining or on frozen sections using β-Gal Staining Kit (#K146501; Invitrogen) by following the manufacturer's instructions. Both preparations gave identical results.

**Reporting Summary**. Further information on research design is available in the Nature Research Reporting Summary linked to this article.

## Data availability
The datasets generated during and/or analyzed during the current study are available from the corresponding author on reasonable request.

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

## Acknowledgements

We would like to thank G.P. Dotto for initial support of the project; M.C. Reymond for assisting SEM; N. Ditzel for technical assistance of animal breeding, and A. Miyawaki for mAG-hGeminin (1/110)-pCSII-EF-MCS and mCherry-hCdt1(30/120)-pCSII-EF-MCS vectors. This work was supported by the National Natural Science Foundation of China to H.Z. (30801289 and 81371138), National Institutes of Health to S.C. (RO1DE019802), the Deutsche Forschungsgemeinschaft (SFB 655 B3) to D.C., the European Union Marie Skłodowska-Curie Actions (618930, OralStem FP7-PEOPLE-2013-CIG), the European Regional Development Fund and the Biotechnology and Biological Sciences Research Council of the UK (BB/L02392X/1) to B.H.

## Author contributions

J.V.W., H.Z., D.S., C.S.I., W.L.K., Y.G., C.B., Y.L., P.R.C.G., C.T., and B.H. performed experiments, collected, and analyzed data. J.K. and D.C. performed FACS analysis. K.K.S. and R.A. provided *PDGFrβ Cre ER*^T2 and *ROSA* ^mT/mG mice. M.Z., K.Y., and Y.B. performed incisor clipping experiment. T.L. and A.C.Z. designed and generated Ki67p-T2A-FUCCI cell cycle reporter. M.Z. and S.A. performed SEM analysis. S.W., Q.L., and J. L. collected clinical samples. M.K. and B.M.A. prepared *Dlk1 knockout* and *Col-Dlk1 Tg* mice. S.C. prepared *mDSPP* promoter constructs. B.H. wrote the manuscript.

## Additional information

**Competing interests:** The authors declare no competing interests.

