## [Peer Review File · Nature Communications]

Reviewers' comments:

Reviewer #1 (Remarks to the Author):

The manuscript by Walker et al. is an ambitious study that spans basic molecular information that may be exploited to demonstrate potential utilization in the clinic. The authors show that a novel subset of mesenchymal stem cells (CL-MSCs), which are located above the epithelial stem cells in the mouse incisor, give rise to mesenchymal transit amplifying cells (MTACs) under the control of Dlk1. Dlk1, which is under epigenetic control affects MSC priming and renewal. Forced expression of Dlk1 in a caries model led to the formation of mineralized osteodentin nodules in the pulp.

This manuscript was very ambitious in its scope and there are several questionable interpretation of data. However, despite this I believe there is something interesting here and I would suggest that this manuscript be considered for publication with major revisions once the comments below are addressed.

Major comments:

- 1) Most experiments are performed in P7 incisors. This is a bit tricky since it is impossible to distinguish between incisor development vs renewal (if in fact there is a difference). What was the justification for looking at P7?
- 2) Authors state, "no evidence has shown that the MTACs are direct derivatives of the NVB-MSCs" and that evidence hint the co-existence of the other MSC population(s)..." I would argue that there is still little evidence presented in this manuscript.
- 3) Results paragraph 2: "SmarcA2 and Zbrtb20 both showed transitional expression overlapping with Ki67...further confirmed CL-MSCs as the progenitors of MTACs." This interpretation is weak. Without lineage tracing using (for example) SmarcA2CreER+reporter, I don't think this conclusion can be made here. Also, how was the MTAC region defined in Fig. 1G? It appears that there is much Ki67+SmarcA2 further distal to the indicated region.
- 4) How was the MTAC region defined and determined? (Figs. 1G, 4F, 5E)
- 5) Results paragraph 2: "NVB structures had a short life of less than 3 days..." How was this determined? Was TUNEL staining performed?
- 6) Results ("CL-MSCs and MTACs....Notch pathway profiles): "Using a transgenic Notch reporter...CL-MSCs particularly exhibited high levels of Notch signaling" – I actually do not observe this NotchICD in Fig 3A.
- 7) Results "CL-MSCs and MTACs....Notch pathway profiles": What is the reason for the discrepancy in RNA and protein differences for Hes1? Has this been reported previously?
- 8) Results "CL-MSCs...Lineages": "LacZ staining specifically labeled some CL-MSCs cells...(Fig. 4A) – This is difficult to see. A magnification of this region would help.
- 9) Results "CL-MSCs...Lineages": "hence confirmed again that CL-MSCs were indeed progenitors of MTAC and odontoblasts." I don't think you can make this conclusion from the data. Again lineage tracing using a tractable mouse model would be greatly helpful. Also, why are DSP and DMP1 seemingly expressed everywhere in the incisor pulp in Fig. 4C with knockout of RBP-Jkappa? Does this mean that all the pulp cells have become odontoblasts? And ameloblasts seem to have high expression of DSP and DMP1. What is going on? Are the images overexposed here?
- 10) Fig. 5E – again, how was the MTAC region determined? It seems quite arbitrary and would depend on the plane of section.
- 11) I do not observe "early differentiation of MTACs and tooth pulp cells...(Figure 5G and H)." Also, the figure is missing "G"
- 12) Fig. 5I – this figure is not informative. Magnifications of the regions of interest are required here. Also, can you somehow quantify the differences in dentin layers?

13) "Therefore, losing Dlk1 could mirror tooth defects of canonical Notch null mice" – I don't think tooth defects of canonical Notch null mice have ever been shown since most of these mice are early lethal.

14) Potential translation work is confusing. Whereas, Dlk1 KO apparently led to premature dentin and dentin repair, in the translation model, the authors seems to suggest that over-expression of Dlk1 leads to reparative dentin. Also, it seems that the osteodentin generated in Fig. 8F is relatively far from the wound. Thus, it appears that pulp stones may have been generated, which would be counter to what we would want to occur and may impede comprehensive treatment.

Minor comments:

1) Introduction 3rd paragraph: "SC-TAC zone at its remote (or distal) end of tooth" is wrong. The cervical loop is on the proximal or posterior end of the incisor.

2) Introduction 3rd paragraph: "cells start polarizing and deposit enamel matrix at the epithelial-mesenchymal junction (Figure 1A and C)." Figure 1A and C should label polarizing cells and enamel matrix.

3) Introduction 3rd paragraph: "TAC (MTAC) zoneexpress Ki67 (Figure 2C)." Do you mean Fig 1C?

4) Fig. 1B – should indicate the plane of view in Fig. 1A.

5) Introduction 4th paragraph: "are are" should be corrected

6) Results 1st paragraph: "that indeed that the" should be changed to "indeed that the"

7) Results 1st paragraph: remove "etc" after "Zbtb20"

8) Results 1st paragraph: "in comparing to NVB-MSCs" should be changed to something like "distinct from NVB-MSCs"

9) Fig. 1F: magnification of CL-MSc region would be helpful

10) Results paragraph 2: "CL-MSCs are progenitors...fully functional odontoblasts." This statement is over-reaching since we do not know that these odontoblasts are fully functional.

Reviewer #2 (Remarks to the Author):

Dear editor,

Dear authors,

I had to review the manuscript by J.V. Walker and colleagues entitled "Transit Amplifying cells coordinate mouse incisor mesenchymal stem cell activation" submitted to Nature Communications.

In this manuscript the authors are using from in vitro to in vivo methods to get a better understanding at a restricted cell population that was called CL-MSCs. The manuscript presented this population as key player in the proper mesenchymal lineage formation, and possibly as target for dental wound healing. While the objective of the study is ambitious, it seems that the authors were not able to provide convincing enough data to achieve their goal. One of the main goals is to demonstrate that the CL-MSCs are stem cells. They did not provide any golden standard experiments.

I have listed here the main issues I have with the current version of the manuscript, and I am sure that the authors will be able to answer to these requests to increase the quality of the manuscript.

- Gene expression should be noted in italics when referred to mRNAs
- Most of the experiments are done at P7. At this stage the incisor can't be considered as adult. Some staining at P30 should be performed
- A large part of the markers expressed in the CL-MSCs are expressed in the epithelial SCs. That should be mentioned. Could Sox2 be used as control in the qPCR experiments to demonstrate the

purity of the mesenchymal cell isolation?

- On the figures, a dotted line should be used to separate epithelium and mesenchyme
- Figure 2B: a picture of the staining before culture should be shown, instead of the small picture showing filaments
- Figure 2C: the section is on the side of the sample, difficult to know if the NVB would be visible here
- Page 7, it is claimed that CL-MSCs contribute to CL growth ex vivo. Where is the data supporting this claim?
- Figure 2D-F: The devitalized dentin slice should be shown to be acellular. How were the CL-MSCs isolated? Was it pre-odontoblast free? What is the white signal on the "merge" picture? What is the negative control here (a dental mesenchymal cell population that would not adopt the odontoblast phenotype)? It would rule out a transdifferentiation process.
- Figure 3B: what is MSC?
- Figure 3D: these pictures are not of good enough quality
- "the Cre gene was shut off at the CL-MSCs stage" I disagree with the statement as it is expressed in some of these cells.
- The DNA deletion part is not good enough. To prove that the CL-MSCs are stem cells, BrdU retaining experiment with long chase and classical genetic fate mapping should be provided: Coll1a2Cre x Rosa26R-LacZ, or other reporter at different time points during several weeks.
- Figure 8B: which are the proteins stained?
- Figure 8C: seeing a healthy area would help in comparing.
- The TACs balancing CL-MSCs part of the discussion is pretty unclear and should be reinforced in the discussion
- Kauka et al: some pictures tend to show that Schwann cell progenitors give rise to the CL-MSCs, could the authors comment this?
- Incomplete materiel and methods part
- The language should be checked

I am really confident that these suggestions will improve the quality of the manuscript and it will meet the standards necessary for acceptance in Nature Communications.

Best regards,

Reviewers' comments:

Reviewer #1 (Remarks to the Author):

The manuscript by Walker et al. is an ambitious study that spans basic molecular information that may be exploited to demonstrate potential utilization in the clinic. The authors show that a novel subset of mesenchymal stem cells (CL-MSCs), which are located above the epithelial stem cells in the mouse incisor, give rise to mesenchymal transit amplifying cells (MTACs) under the control of Dlk1. Dlk1, which is under epigenetic control affects MSC priming and renewal. Forced expression of Dlk1 in a caries model led to the formation of mineralized osteodentin nodules in the pulp.

This manuscript was very ambitious in its scope and there are several questionable interpretation of data. However, despite this I believe there is something interesting here and I would suggest that this manuscript be considered for publication with major revisions once the comments below are addressed.

Answer from the authors:

We thank the reviewer's very detailed comments that indeed helped us to improve the quality of the work. The manuscript reflects a major project of our group for the past 10 years. Hence it covers many aspects. We have extensively revised the manuscript based on the reviewer's comments:

- (1) Incorporating a new inducible MSCs lineage tracing model: *PDGFRb* *ER*^{T2} x *ROSA*^{mT/mG} mice, which allows tracing mesenchymal progenitors in the incisor;
- (2) Adding additional results that support our conclusions;
- (3) Changing the manuscript according to the reviewer's comments.

Please see below point-by-point answers.

Major comments:

1) Most experiments are performed in P7 incisors. This is a bit tricky since it is impossible to distinguish between incisor development vs renewal (if in fact there is a difference). What was the justification for looking at P7?

Answer from the authors:

When we started the project, we indeed compared new born (P7) with adult mice (P30) for the expression of the markers we identified and we confirmed that the majority of the MSCs markers at P7 still persist in P30 hence we did not see significant difference. please see Fig. 3e-h, and Supplementary Fig. 4a and c. Our markers mainly work on freshly frozen tissues therefore P7 is more practical as we can cut the whole head and analyse the entire cervical loop from labial-lingual direction for each marker. For P30 we have to dissect fresh cervical loop from the mineralised mandible before it can be frozen

and processed. Consequently, the morphology of the cervical loop is difficult to preserve hence successful rate is low. Therefore to minimise mouse usage initially we decided to focus on P7.

2) *Authors state, “no evidence has shown that the MTACs are direct derivatives of the NVB-MSCs” and that evidence hint the co-existence of the other MSC population(s)...” I would argue that there is still little evidence presented in this manuscript.*

Answer from the authors:

We have revised the manuscript to include the lineage tracing experiments performed with the *PDGFrβ* *ER^{T2}* x *ROSA^{mT/mG}* mice that we received helps from Professor Ralf Adams, who developed the *PDGFrβ* *ER^{T2}* mice. We now confirm that the cervical loop MSCs are the major source for MTACs both for P7 and P30 (new Fig.3). However, we agree that we cannot exclude that NVB-MSCs also contribute to MTACs, hence we changed the text into “little evidence...” (page 6, line 10 from the bottom).

3) *Results paragraph 2: “SmarcA2 and Zbrtb20 both showed transitional expression overlapping with Ki67...further confirmed CL-MSCs as the progenitors of MTACs.” This interpretation is weak. Without lineage tracing using (for example) SmarcA2CreER+reporter, I don't think this conclusion can be made here.*

Answer from the authors:

Currently there are no *SmarcA2 Cre ER* mouse lines available. Instead, we now included the *PDGFrβ* *ER^{T2}* x *ROSA^{mT/mG}* mice, as mentioned above, which enabled us to label the MSCs in the incisor and the results clearly showed that the CLE-MSCs is the major progenitor source for MTACs (new Fig. 3).

Also, how was the MTAC region defined in Fig. 1G? It appears that there is much Ki67+SmarcA2 further distal to the indicated region.

Answer from the authors:

The indicated region in Fig.1g is the transition region between CL-MSCs and MTACs, which hence shows overlapped SmarcA2 with Ki67 signals.

4) *How was the MTAC region defined and determined? (Figs. 1G, 4F, 5E)*

Answer from the authors:

For Fig. 1g, as mentioned above, the square region represents the transition of CL-MSCs and MTACs.

For the old Fig. 4f, 5e, due to quantification purpose, we had to find a way which we could arbitrarily define a region to compare the cells that were positive for certain markers, particularly Ki67. Unlike the incisor epithelium (which we published recently, Singer et. al., the EMBO Journal, DOI 10.15252/embj.201899845), in the mesenchyme, while the Ki67 positive cells concentrate near the epithelial-mesenchyme junction and neighbour the epithelial transit amplifying cells, there are still diffuse positive cells which reside within the tooth pulp more distantly from CL adjacent area. Therefore, we arbitrarily defined the region by drawing a straight line between the epithelial tip, where marks boundary the epithelial stem cells and transit amplifying cells (Singer et. al., the EMBO Journal, DOI 10.15252/embj.201899845) with the first polarized preodontoblasts, and measured the cells included in the region as the studying target. We detailed these methods in the revised manuscript (page 36-37) The quantification results represent the Ki67 positive cells in that region on serial sections. For the old Fig. 4F (the new Fig.5e) the drawing of the region was not correct, we have now replaced with the right ones.

5) *Results paragraph 2: “NVB structures had a short life of less than 3 days...” How was this determined? Was TUNEL staining performed?*

Answer from the authors:

The NVB structures were evaluated by immunofluorescent staining for CD106 and Neural Filament which mark NVBs in the incisor tooth (new Fig. 2b, c). For the cultured explants, we performed serial sections and could not see obvious CD106 and Neural Filament staining (new Fig. 2c). With TUNEL labelling redone on the same set of samples, we can see there are many apoptotic cells particularly at the periphery and some inside the explants (new Fig. 2c). This potentially explains the reason for the disappearance of the endothelial cells.

6) *Results (“CL-MSCs and MTACs....Notch pathway profiles): “Using a transgenic Notch reporter...CL-MSCs particularly exhibited high levels of Notch signaling” – I actually do not observe this NotchICD in Fig 3A.*

Answer from the authors:

Notch1 ICD is the cleaved form of Notch receptor and its translocation into the nuclei represents the activation of the Notch pathway. In Fig. 3A (new Fig. 4A), we did double staining of EGFP (for the reporter) and Notch1 ICD and the nuclear signal of Notch1 ICD is evident.

7) *Results “CL-MSCs and MTACs....Notch pathway profiles”:* *What is the reason for the discrepancy in RNA and protein differences for Hes1? Has this been reported previously?*

Answer from the authors:

For some molecules, it can take in excess of 24 hours from mRNA expression to protein production, hence mRNA and protein expressions are not necessarily the same. Also, the post-translational stability varies between molecules. While we believe the high mRNA expression of Hes1, as well as Hes5, Hey1, Hey2 and Heyl represent the activation of canonical Notch pathway, we don't know yet the mechanism of delayed response of Hes1 protein production which might be due to feedback inhibition, transcription delay or translation delay. The delayed Hes1 protein production has been described and modelled previously (e.g. Barrio et al., doi: [10.1371/journal.pcbi.0020117](https://doi.org/10.1371/journal.pcbi.0020117) and Tzou et al., doi: [10.1089/cmb.2014.0022](https://doi.org/10.1089/cmb.2014.0022)). However, we believe this does not affect the interpretation of our results.

8) Results “CL-MSCs...Lineages”: “LacZ staining specifically labeled some CL-MSCs cells... (Fig. 4A) – This is difficult to see. A magnification of this region would help.

Answer from the authors:

We now include the magnified image of the region (new Fig. 5a to better illustrate the LacZ positive cells).

9) Results “CL-MSCs...Lineages”: “hence confirmed again that CL-MSCs were indeed progenitors of MTAC and odontoblasts.” I don't think you can make this conclusion from the data. Again lineage tracing using a tractable mouse model would be greatly helpful.

Answer from the authors:

Please see our answers for question 2) and 3) for the new lineage tracing model.

Also, why are DSP and DMP1 seemingly expressed everywhere in the incisor pulp in Fig. 4C with knockout of RBP-Jkappa? Does this mean that all the pulp cells have become odontoblasts? And ameloblasts seem to have high expression of DSP and DMP1. What is going on? Are the images overexposed here?

Answer from the authors:

In the *RBP-Jkappa* knockout mice, incisor pulp cells started to express DSP and DMP1 (new Fig. 5B, old Fig. 4c), very possibly due to the loss of this transcriptional factor from the binding of the promoter of DSPP and DMP1 (new Fig. 5f). However, the expression of DSP and DMP1 in pulp cells did not necessarily indicate that the cells become normal odontoblasts. Instead, the cells might undergo an abnormal differentiation program and secrete irregular matrix etc.

We confirm the image exposure are the same between control and knockout samples. We did observe high expression of DSP and DMP1 also in the epithelial compartment, which we believe is true. But our paper focused on mesenchymal cells and the starting expression of DSP and DMP1 in the epithelial cells in the RBP-Jkappa mice might be interesting for future detailed works studying the mesenchymal-epithelial interaction.

10) Fig. 5E – again, how was the MTAC region determined? It seems quite arbitrary and would depend on the plane of section.

Answer from the authors:

Please see answers to question 4). After trying different methods, we adopted the reported protocol as it could facilitate the analysis on the complete sets of serial sections of the cervical loops of the indicated mouse strains used in the study.

11) I do not observe “early differentiation of MTACs and tooth pulp cells... (Figure 5G and H).” Also, the figure is missing “G”

Answer from the authors:

The “early differentiation of MTACs and tooth pulp cells...” referred to increased DSPP/DSP and DMP1 expression in the tooth pulp of the *Dlk1* KO mice. However, we agree that it might not be entire “early differentiation”, so we have rephrased the sentence (page 12, line 6 from the bottom).

“G” was on the original figure.

12) Fig. 5I – this figure is not informative. Magnifications of the regions of interest are required here. Also, can you somehow quantify the differences in dentin layers?

Answer from the authors:

We regret the images are not very clear because since even the original micro-CT scan was done at high resolution, the orientation of the mouse heads was not very sagittal therefore we had to use Imaris software to “resection” the micro-CT image. As suggested by the reviewer, now we provide quantitative data for the dentin layer volume analysis to support our results (Fig. 6j).

*13) “Therefore, losing *Dlk1* could mirror tooth defects of canonical Notch null mice” – I don't think tooth defects of canonical Notch null mice have ever been shown since most of these mice are early lethal.*

Answer from the authors:

Here we referred to the RBP-Jkappa conditional knockout mice that we have used in the study as the molecule is the key transcriptional factor of Notch pathway.

14) Potential translation work is confusing. Whereas, Dlk1 KO apparently led to premature dentin and dentin repair, in the translation model, the authors seems to suggest that over-expression of Dlk1 leads to reparative dentin. Also, it seems that the osteodentin generated in Fig. 8F is relatively far from the wound. Thus, it appears that pulp stones may have been generated, which would be counter to what we would want to occur and may impede comprehensive treatment.

Answer from the authors:

In our report, we found that Dlk1 is one important factor in the transit amplifying cells for balancing stem cell preservation and differentiation: the Dlk1 KO mice have premature or enhanced dentin formation but reduced stem cell pool, and Dlk1 overexpression mice have enlarged the stem cell pool and accelerated cell differentiation and reparative dentin like mineralization instead. Translation application of our findings require a good strategy to preserve stem cells and at the same time enhance/accelerate tissue mineralization. Beside the molar tooth pulp capping model, we now also include an incisor clipping experiment, where we show stimulating wound healing does increase Dlk1 expression in the MTACs (Supplementary Fig. 4b, c). Hence applying Dlk1 on the tooth pulp appeared to be a potential solution for translational application, as we found in the study.

“Pulp stone” formation in normal tooth represents excessive differentiation of tooth pulp cells and subsequent mineralization, often tertiary dentin. In normal tooth, the “pulp stone” can impede pulp treatment. In caries, it might represent active reparative processes of the pulp cells instead. And treatment on caries desire new dentin like structures’ quick formation and mineralization. Our results showed that in human caries Dlk1-Notch2 links persist in the reparative odontoblasts, as well as cells around the “pulp stone”, suggesting Dlk1 could be used for potential clinical application. Concerning the rat capping experiments, for the newly formed dentin like structures near the opening of the root pulp, we have consulted Prof. Paul Cooper from University of Birmingham, UK, the expert of dentin research, who confirmed those structures are reparative dentin. However, we agree the current study is preliminary, and the application of Dlk1 on tooth capping needs to be further tested on large animals as the rat model is limited by the tooth size.

Minor comments:

1) Introduction 3rd paragraph: “:SC-TAC zone at its remote (or distal) end of tooth” is wrong. The cervical loop is on the proximal or posterior end of the incisor.

Answer from the authors:

This now has been corrected as “posterior”.

2) *Introduction 3rd paragraph: “cells start polarizing and deposit enamel matrix at the epithelial-mesenchymal junction (Figure 1A and C).” Figure 1A and C should label polarizing cells and enamel matrix.*

Answer from the authors:

This now has been added on Fig. 1a for dentin and Fig. 1c for polarized preodontoblasts.

3) *Introduction 3rd paragraph: “TAC (MTAC) zoneexpress Ki67 (Figure 2C).” Do you mean Fig 1C?*

Answer from the authors:

This now has been corrected as Fig. 1c.

4) *Fig. 1B – should indicate the plane of view in Fig. 1A.*

Answer from the authors:

This now has been added on Fig. 1b for the plane direction.

5) *Introduction 4th paragraph: “are are” should be corrected*

Answer from the authors:

This now has been corrected.

6) *Results 1st paragraph: “that indeed that the” should be changed to “indeed that the”*

Answer from the authors:

This now has been corrected.

7) *Results 1st paragraph: remove “etc” after “Zbtb20”*

Answer from the authors:

This now has been corrected.

8) *Results 1st paragraph: “in comparing to NVB-MSCs” should be changed to something like “distinct from NVB-MSCs”*

Answer from the authors:

This now has been changed to “Conversely, NVB-MSCs expressed...”

9) *Fig. 1F: magnification of CL-MSc region would be helpful*

Answer from the authors:

We believe under the current magnification the signals are clear and we have limited space to insert magnified CL region. However, we are happy to do that if the reviewer insist.

10) *Results paragraph 2: “CL-MSCs are progenitors...fully functional odontoblasts.” This statement is over-reaching since we do not know that these odontoblasts are fully functional.*

Answer from the authors:

The sentence has been removed, as suggested.

Reviewer #2 (Remarks to the Author):

Dear editor,

Dear authors,

I had to review the manuscript by J.V. Walker and colleagues entitled “Transit Amplifying cells coordinate mouse incisor mesenchymal stem cell activation” submitted to Nature Communications.

In this manuscript the authors are using from in vitro to in vivo methods to get a better understanding at a restricted cell population that was called CL-MSCs. The manuscript presented this population as key player in the proper mesenchymal lineage formation, and possibly as target for dental wound healing. While the objective of the study is ambitious, it seems that the authors were not able to provide convincing enough data to achieve their goal. One of the main goals is to demonstrate that the CL-MSCs are stem cells. They did not provide any golden standard experiments.

I have listed here the main issues I have with the current version of the manuscript, and I am sure that the authors will be able to answer to these requests to increase the quality of the manuscript.

Answer from the authors:

We would like to thank the reviewer for the positive and constructive comments on our work. In order to better characterise the CL-MSCs stem cells, beside the new inducible MSCs lineage tracing model: *PDGFrβ* *ER*^{T2} x *ROSA*^{mT/mG} mice (Fig. 3) which showed CL-MSCs cells do give rise to odontoblast, we have also performed additional tests and confirmed that the CL-MSCs could differentiate into adipocytes, osteoblasts and chondrocytes (Supplementary Fig. 3).

We have performed additional experiments and changes based on the reviewer’s suggestions. Please see below point-to-point answers.

- Gene expression should be noted in italics when referred to mRNAs

Answer from the authors:

This now has been corrected.

- Most of the experiments are done at P7. At this stage the incisor can’t be considered as adult. Some staining at P30 should be performed

Answer from the authors:

Please see our reply to Reviewer 1's question 1), we indeed have performed P30 experiments and please find the corresponding results.

- A large part of the markers expressed in the CL-MSCs are expressed in the epithelial SCs. That should be mentioned. Could Sox2 be used as control in the qPCR experiments to demonstrate the purity of the mesenchymal cell isolation?

Answer from the authors:

We now have mentioned that the CL-MSCs markers also expressed in the epithelial SCs. [Redacted]

- On the figures, a dotted line should be used to separate epithelium and mesenchyme

Answer from the authors:

We have added dotted lines on all the corresponding images for illustrating the epithelial-mesenchymal junction.

- Figure 2B: a picture of the staining before culture should be shown, instead of the small picture showing filaments

Answer from the authors:

This has been now added (Fig. 2b)

- Figure 2C: the section is on the side of the sample, difficult to know if the NVB would be visible here

Answer from the authors:

Please also see our answer to Reviewer 1's comment 5) (Page3 of this document), we have analysed each explant on serial sections and confirmed the results. Please also see the new sample and result (Fig. 2c).

- Page 7, it is claimed that CL-MSCs contribute to CL growth ex vivo. Where is the data supporting this claim?

Answer from the authors:

We referred to Fig. 2a-e, where in the absence of NVB, the CL region could still develop and generate new tissues. In Fig. 2a it is clear that comparing with the initial freshly isolated tissue (the new panel we have included), the explant has grown a lot.

- Figure 2D-F: The devitalized dentin slice should be shown to be acellular. How were the CL-MSCs isolated? Was it pre-odontoblast free? What is the white signal on the “merge” picture? What is the negative control here (a dental mesenchymal cell population that would not adopt the odontoblast phenotype)? It would rule out a transdifferentiation process.

Answer from the authors:

We now added the devitalized dentin slice images (Fig. 2h). The CL-MSCs were isolated by microdissecting CL first then further dissociated using Dispase II (middle of page 24), the dissection was performed by Dr Bing Hu. We have included negative control (the tooth pulp) as well (Fig. 2j), where although DMP1 and DSP expression could be observed, no cell protrusions could be identified.

- Figure 3B: what is MSC?

Answer from the authors:

It should be CL-MSCs. It has been now corrected.

- Figure 3D: these pictures are not of good enough quality

Answer from the authors:

In fact, those pictures were overlapped with phase contrast images to better illustrate dentin. We have now removed the phase contrast to better show the immunofluorescence (new Fig. 4d).

- “the Cre gene was shut off at the CL-MSCs stage” I disagree with the statement as it is expressed in some of these cells.

Answer from the authors:

We agree with the reviewer and have removed the sentence.

- The DNA deletion part is not good enough. To prove that the CL-MSCs are stem cells, BrdU retaining experiment with long chase and classical genetic fate mapping should be provided: Coll1a2Cre x Rosa26R-LacZ, or other reporter at different time points during several weeks.

Answer from the authors:

We have removed DNA deletion and included the *PDGFrβ* ERT2 x *ROSA*^{mT/mG} mice, as mentioned above and showed in Fig. 3.

- *Figure 8B: which are the proteins stained?*

Answer from the authors:

Sorry for missing the labelling. The green signal represents Dlk1 and red signal represents Notch2. We have added the labelling on the panels.

- *Figure 8C: seeing a healthy area would help in comparing.*

We now included a relative healthy region of the same tooth (new Fig. 9b)

- *The TACs balancing CL-MSCs part of the discussion is pretty unclear and should be reinforced in the discussion*

Answer from the authors:

We have rewritten the discussion, particularly the TACs balancing CL-MSCs part (page 19).

- *Kauka et al: some pictures tend to show that Schwann cell progenitors give rise to the CL-MSCs, could the authors comment this?*

Answer from the authors:

The works from Kauka et al. showed Schwann cell progenitors can give rise to incisor tooth pulp cells including odontoblasts and pulp cells. However, it is not clear if the MTACs and CL-MSCs are also positive. As although from Fig. 1d and i, it looks like the MTACs and CL-MSCs are positive for the reporter, however based on the images, we are not convinced that the drawing of the line is entirely correct for the epithelial CL outlines. Please see below image indicating what we consider as where are the real shape and outline of the CL (based on Kauka et al., Figure 1d and i, images were turned 90-degree, we believe the round dot lines in cyan mark the real CL outlines). It would be nice if the authors of that paper can clarify the situation by providing high resolution images. And for their Fig. 3d, we believe the marked CL might not at the stem cell containing region based on the image and drawing. Again, it will be clear if high magnification images are provided.

Concerning the Schwann cell origin, Kauka's report highly relied on the *PLP-Cre ER^{T2}* transgenic line, which they claimed uniquely labelled Schwann cell progenitors. However, it is our concern that there is increasing evidence and debate that the same Cre also labels neural crest cells. The issue has been recently extensively discussed by Prof. Lukas Sommer in 2018 (DOI: 10.1002/dvg.23105, page 6-7). Therefore, it is still possible the report showed, transiently expressed PLP Cre labelled cells are a fraction of embryonic neural crest cells. To validate if Schwann cell progenitors do contribute to odontoblasts etc., it would be interesting to repeat the experiments with the same and different other Cre line(s) by an independent group to see how to differentiate the different Cre transgenic mouse lines and compare the populations labelled. We also discussed about the points now in Discussion section (middle of page 18).

- Incomplete material and methods part

Answer from the authors:

We have rechecked material and methods section and have added the missing information.

- The language should be checked

Answer from the authors:

We have rechecked language and invited colleagues of native English speakers to proof read the manuscript.

REVIEWERS' COMMENTS:

Reviewer #1 (Remarks to the Author):

I am quite satisfied with the revisions. Thank you.

Reviewer #2 (Remarks to the Author):

Dear editor,

Dear authors,

I had to review the revised version of the manuscript by J.V. Walker and colleagues entitled "Transit Amplifying cells coordinate mouse incisor mesenchymal stem cell activation" submitted to Nature Communications.

After a careful review of how the authors revised their manuscript, and the large amount of new data presented, I am confident that this study is now more comprehensive, and the data truly support the statements made in the text.

This work is not only of interest for the dental specialists, but as well of interest for the developmental biology and regenerative medicine.

Therefore, I recommend the manuscript to be accepted for publication in Nature Communications.

Best wishes,

REVIEWERS' COMMENTS:

Reviewer #1 (Remarks to the Author):

I am quite satisfied with the revisions. Thank you.

Reviewer #2 (Remarks to the Author):

Dear editor,

Dear authors,

I had to review the revised version of the manuscript by J.V. Walker and colleagues entitled “Transit Amplifying cells coordinate mouse incisor mesenchymal stem cell activation” submitted to Nature Communications.

After a careful review of how the authors revised their manuscript, and the large amount of new data presented, I am confident that this study is now more comprehensive, and the data truly support the statements made in the text.

This work is not only of interest for the dental specialists, but as well of interest for the developmental biology and regenerative medicine.

Therefore, I recommend the manuscript to be accepted for publication in Nature Communications.

Best wishes,

ANSWER FROM THE AUTHORS:

We appreciate both reviewers' time and generous helps to highly improve our manuscript.